# Magnetic local time asymmetries in precipitating electron and proton populations with and without substorm activity

Olesya Yakovchuk[1,2] and Jan Maik Wissing [1]

[1]The Institute of Environmental Systems Research, University of Osnabrück, Osnabrück, 49069, Germany
[2]Skobeltsyn Institute of Nuclear Physics, Lomonosov Moscow State University, Moscow 119234, Russia

**Correspondence:** Olesya Yakovchuk (oyakovchuk@uos.de)

**Abstract.** The magnetic local time (MLT) dependence of electron (0.15–300 keV) and proton (0.15–6900 keV) precipitation into the atmosphere based on National Oceanic and Atmospheric Administration POES and METOP satellites data during 2001–2008 was described. Using modified APEX coordinates the influence of particle energy, substorm activity and geomagnetic disturbance on the MLT flux distribution was statistically analysed.

Some of the findings are: a) Substorms mostly increase particle precipitation in the night-sector by about factor 2–4 but can also reduce it in the day-sector. b) MLT dependence can be assigned to particles entering the magnetosphere at the cusp region and magnetospheric particles in combination with energy-specific drifts (in agreement with Newell et al., 2009). c) MLT flux differences of up to two orders of magnitude have been identified inside the auroral oval during geomagnetically disturbed conditions. Novelty here is the comprehensive coverage of energy bands and the focus on asymmetry. d) The maximum flux asymmetry ratio depends on particle energy, decreasing with Kp for low energetic particles and increasing with Kp for higher energy electrons, while high energy protons show a more complex dependency. While some aspects may already been known, the quantification of the flux asymmetry sheds new light on MLT variation.

## 1  Introduction

Particle precipitation is a primary link between solar activity and atmospheric chemistry. Thorne (1977) suggested a depletion of Ozone in 40–80 km through production of nitric oxides by precipitation of relativistic radiation belt electrons. Ozone depletion following solar energetic particle events (mostly protons) has been observed in the same year (Heath et al., 1977). Auroral particle precipitation causes production of $HO_y$ and $NO_x$ and thus is a significant player in mesospheric and stratospheric chemistry, especially as these chemicals catalytically impact the ozone cycle (Callis et al., 1996b, a) and subsequently change the radiation budget and affect dynamics. Consequently there has been an immanent need for the description (and later on modelling) of the particle precipitation. And even though the investigation of precipitation pattern of low energetic particles (and especially electrons) started more then 30 years ago (Hardy et al., 1985, e.g.), the rising vertical extend of climate models has shifted the focus from high energetic particles to lower energies again.

The interplanetary medium is the driver of geomagnetic disturbance and may compress, deform or reconnect to the magnetosphere. Meredith et al. (2011) e.g. states that on average, the flux of precipitating energy electrons (E>30 keV) is enhanced

by a factor of about 10 during the passage of the high-speed stream (geomagnetic storm time) at all geographic longitudes. Thus geomagnetic disturbance should be considered in a description of particle precipitation.

MLT dependence is a result of charge-dependent drift directions (Allison et al., 2017) and (linked to that) opposite potentials in field-aligned Birkeland-currents. The authors themselves note that the particle flux variety in different local time sectors may
reach an order of magnitude, with proton precipitation dominating in evening and night sectors and electrons dominating in morning and night (Wissing et al., 2008).

Substorms are either directly driven or/and loading processes, where energy is accumulated and released abruptly in the Earth's magnetosphere (Akasofu, 2015). The global morphology of auroral substorms has first been described by Akasofu (1964) using simultaneous all-sky camera recordings from Siberia, Alaska and Canada. Later space-born missions like e.g. the
UV photometer mission on Dynamics Explorer 1 (DE-1) (Frank et al., 1981) confirmed this morphology.

Akasofu et al. (1965) also already characterized the expansion phase and the recovery phase of a substorm (a preceding growth phase has been added by McPherron (1970)). Due to auroral emissions the substorms were associated with excitation and ionization by precipitating particles that have been investigated by ground based riometers (Berkey et al., 1974) and later on by satellite missions (e.g. Fujii et al., 1994), observing intense energetic electron precipitation in or near the onset/surge
region. The energy range of the precipitating particles has been defined as electrons and protons at approx. 10–100 keV with a low-energy cut-off (Birn et al., 1997). The precipitation regions depend on particle species.

The occurrence of substorms depends on the orientation of the interplanetary magnetic field. As shown in Reeves et al. (2003) these external solar wind parameters subsequently impact the magnetic field on the ground and are represented in the Auroral Electrojet (AE) index. Auroral Electrojet indices AE=AU-AL are a good proxy of the global auroral power, where AU
and AL are the upper and lower components of AE, which means the largest and smallest values of the H component among 12 magnetic stations (Davis and Sugiura, 1966). AU represents the strength of the eastward electrojet, while AL represents the westward electrojet. Consequently AL seems to be the index which best corresponds to westward intensification of the auroral current aka substorm activity. Prior to substorm onset, AL index is typically small in magnitude, with the contributing station near dawn, whereas during substorm onset, the station contributing to the lower envelope is usually in the dusk sector under the
auroral expansion. However, due to the limited spatial coverage of the 12 magnetometer stations the auroral expansion can be missed, which means that this index does not always reflect the onset (Gjerloev et al., 2004). The use of SuperMAG SML, an index derived likewise to the AE but based on all available magnetometer stations (typically more than 100) at these latitudes, considerably improves the detection of substorm onsets (Newell and Gjerloev, 2011a). Thus we use the SML index in this study to define substorm onsets.
In this study we will discuss MLT differences in particle fluxes (and precipitation) over a wide energy range and show how substorms impact this pattern.

In Section 2 particle data, modified APEX coordinate system, SML and Kp-binning will be introduced. Section 3 displays the application of modified APEX coordiantes to the flux maps, discusses special aspects of the MLT binning and illustrates how the auroral oval is fitted. In Section 4 the main discussion follows, the analysis of particle fluxes on high latitudes. The
results are summarized in Section 5.

## 2 Data sets

This section describes the data sets and how the data has been processed.

### 2.1 Particle data

For particle data we use time series (2001–2008) of 16 s averaged electron fluxes ranging from 0.15 to 300 keV and protons from 0.15 to 6900 keV measured on board the polar orbiting NOAA/POES and their successor, the METOP satellites (Evans and Greer, 2006). 2001 to 2008 covers the complete declining phase of solar cycle 23 and thus includes very active (sometimes extreme) to very low activity periods (Logachev et al., 2016).

In total all available data from POES 15, 16, 17, 18 and METOP 02 has been used, except for POES 16 after 2006 as it is known that the TED data is erroneous.

All satellites have sun-synchronous orbits at altitudes around 820 km (with $\approx 100$ minute periods of revolution) and an inclination of $\approx 98.5°$. The satellites have initially been placed in orbits that cross the equator in a fixed local time either being morning-evening or day-night sector. However, or in our case fortunately, these orbits were drifting slightly with time. Thus our long sample period and the moving five satellites allowed us to investigate the effect of local time on particle fluxes.

Information about the different channels can be found in Tab. 1. All particle count rates have been converted into differential flux by dividing the energy range and a geometric factor has been applied as suggested in Evans and Greer (2006).

It is known that there is no adequate upper energy threshold of the three MEPED electron channels (Yando et al., 2011). In order to work with specific energy bands we subtracted sequent channels, resulting in the two channels mep0e1-e2 and mep0e2-e3 with the energy bounds given in Tab. 1.

We used the $0°$ detectors only. While the TED $0°$-detector looks exactly radially outward the MEPED $0°$-detector is slightly shifted by $9°$ to ensure a clear field of view (Evans and Greer, 2006).

The MEPED detectors have a field of view of $\pm 15$ degrees, while the TED detector has the following specifications according to Evans and Greer (2006): The field of view of the electron and proton 1000–20 000 eV detector systems are $1.5°$ by $9°$, half angles. The field of view of the 50–1000 eV electron detector system is $6.7°$ by $3.3°$, half angles. The field of view of the 50–1000 eV proton detector system is $6.6°$ by $8.7°$, half angles. Opening angles of the detector in combination with the position of the satellite determines which particle populations the detector is measuring. According to Rodger et al. (2010, Fig. 1) the MEPED 0-degree detector in latitudes discussed in Section 4 measures particles in the bounce loss cone only.

Given that the point of view of the TED detector is almost identical with the MEPED detector and the field of view is significantly smaller, Figure 1 in Rodger et al. (2010) can also be applied, keeping in mind that regions of overlapping particle populations will decline. Thus we can borrow the particle populations seen in the TED channels from the MEPED results. In sum: at high latitudes both detectors count precipitating particle flux while they detect mostly trapped particles at low latitudes.

All figures in this paper are showing differential particle flux in $1/(MeV \, m^2 \, s \, sr)$ as measured, thus we made no assumption about a pitch angle distribution here. However, it should be noted that even if the detector is looking upward (and measuring downgoing particles in high latitudes) it does not necessarily mean that all these particles are precipitating (reaching the

**Table 1.** Channels and nominal energy ranges from the POES and METOP satellites which have been used.

| | instrument | channel | energy range |
|---|---|---|---|
| electrons | TED | band 4 | 154–224 eV |
| | | band 8 | 688–1000 eV |
| | | band 11 | 2.115–3.075 keV |
| | | band 14 | 6.503–9.457 keV |
| | MEPED | mep0e1-e2 | 30–100 keV |
| | | mep0e2-e3 | 100–300 keV |
| protons | TED | band 4 | 154–224 eV |
| | | band 8 | 688–1000 eV |
| | | band 11 | 2.115–3.075 keV |
| | | band 14 | 6.503–9.457 keV |
| | MEPED | mep0P1 | 30–80 keV |
| | | mep0P2 | 80–240 keV |
| | | mep0P3 | 240–800 keV |
| | | mep0P4 | 0.8–2.5 MeV |
| | | mep0P5 | 2.5–6.9 MeV |

atmosphere). Given that some particles are mirroring above the atmosphere a fraction of the downgoing flux is lost, thus the magnetic flux tube is narrowing the particle flux increases again and only in case that the pitch angle distribution is isotrotropic the mirrored fraction is balanced by flux tube narrowing (see e.g. Bornebusch et al., 2010). And only in case of an isotropic pitch

angle distribution it does not matter for upscaling which angles of the downgoing pitch angle distribution we are measuring: an isotropic pitch angle distribution may easily be integrated over $2\pi$ to estimate the total precipitating flux over all angles.

However, it is known that anisotropic distributions occur. While an unaccelerated source population is assumed to be isotropic (as is a wave-scattered fraction of that population in the loss cone) most acceleration processes are connected with an anisotropic pitch angle distribution. Dombeck et al. (2018) lists the most important ones as quasi-static-potential-structures, namely an electric potential field, that may cause isotropic or anisotropic distributions and Alfvén waves, that accelerate only particle energies that are in resonance with magnetic field wave and causes highly anisotropic distributions. Alfvén waves are responsible for electron precipitation during substorms (Newell et al., 2010). According to Newell et al. (2009) electrons are often accelerated while ions are not.

In case of an anisotropic pitch angle distribution an estimation of the total precipitating flux is not straight forward as first a pitch angle distribution has to be assumed and second it has to be determined which pitch angles the detector is currently measuring. Since the only other detector orientation on POES is measuring trapped particles (at high latitudes) and since trapped particles do not get lost, there is no reason to assume a smooth transition between these two particle populations. Thus we do not have a "reference" anisotropic pitch angle distribution that might be applied. Applying an isotropic pitch angle (which is often done in literature) will put the downgoing flux on a level with precipitating flux. In case that the paper states "particle precipitation" this isotropic pitch angle distribution has been implicitly assumed. Yet, this has been made without loss of generality since the shown differential flux in that case is equal to the downgoing flux. Thus no transformation is needed.

All shown values are spatial and temporal averaged fluxes. In case that a detector measures zero counts every time it crosses a specific position and at a certain condition this also enters the figures with zero flux (see e.g. Fig. 2, TED electron band 11, isolated substorm, -55 degrees modified APEX latitude at noon). Since the detector counts are transferred into flux the MEPED channels do not recognize flux less than 1 count per integration interval (equivalent to $1\,000\,000$ particles$/(m^2 ssr)$, divided by the channels energy range). For the TED detector the tranformation is similar but instead of a fixed number a calibration factor has to be applied for every channel and satellite. The calibrations are given in e.g. Evans and Greer (2004).

The particle detectors suffer from various contamination effects: The MEPED electron channels are highly efficient detectors for high energetic protons. In order to avoid contaminated electron data we excluded MEPED electrons when the omnidirectional proton channel P7 showed more than 2 counts (based on high resolution 2 s data). This does not only cut out probably contaminated periods in SPEs, but also the region of the SAA. The MEPED electron channels have been substracted from each other, resulting in differential channels.

Note that the given energy ranges taken from Evans and Greer (2006) are nominal. Some channels suffer from degradation. This mostly holds for the MEPED proton channels and is a result of structural defects caused by the impinging particles. On the long run it causes an energy shift (to higher particles energies) since less electron-hole pairs are produced per deposited particle energy. Consequently the energy ranges mentioned are nominal ranges. Further details on degradation of the MEPED channels can be found in e.g. Asikainen et al. (2012).

## 2.2 Coordinate system

A meaningful representation of particle precipitation has high requirements for the coordinate system as they are: a) The flux pattern should be invariant in time even though the magnetic field is changing (meaning moving poles, not magnetospheric distortion). This is needed for the long investigation period as well as for durability of forecasts. b) The latitude of particle flux pattern should be invariant of the longitude. Given this criterion the longitude may be replaced by local time as second coordinate. c) If the previous criterion is applied, it includes that particle flux has to be recalculated. Following the footpoints of two shells with a distinct magnetic field strength, their latitudinal distance differs with longitude. Since the particle flux is measured on a fixed detector size this has to be taken into account when removing the longitudinal dependence. d) Particle measurements take place at the position of the satellite, which is in about 820 km above the ground. But the effect of particle precipitation (the atmospheric ionization) is mainly located at about 110 km altitude (maximum of magnetospheric ionization, higher particle energies cause ionization futher down). Consequently a coordinate system that allocates the satellite's measurement to their respective position at 110 km altitude would be helpful.

The coordinate system that allows for all named requirements is the modified APEX coordinate system (Richmond, 1995). The coordinates are variable in time using the International Geomagnetic Reference Field model magnetic field configuration, which means they also reflect the temporal movement of the poles. Richmond (1995) present three coordinate systems which are closely connected. The quasi-dipole (QD) coordinates present the magnetic latitude and longitude on the ground (Richmond, 1995, see f1 and f2 base vectors in Fig. 1), while the third base vector goes radially outward. The APEX coordinate system is using the same longitude, but the latitude is following the magnetic field lines as propagating (precipitating) particles do, meaning that a charged particle is always on the same latitude. The APEX latitude is defined by its footpoint on the QD latitude on the surface. In the *modified* APEX coordinates not the surface but an arbitary altitude is used for the definition of the latitude, e.g. that altitude where particles cause the ionization, in our case 110 km above the ground. Thus the measurements should be mapped down on the according field line until it reaches the altitude where the particle is stopped by the atmosphere (about 110 km). In all (modified) APEX systems measurements and ionization location are on the same latitude. Thus a desirable coordinate system for our work is the modified APEX system.

## 2.3 SML index and derived substorm onsets

The period 2001–2008 was chosen for our investigation, where all necessary data about substorms and particle fluxes are available. For the identification of substorm events, we use the technique published by Newell and Gjerloev (2011a). The substorm onset is determined by the auroral electrojet SML index, which is derived from magnetometer data obtained by the SuperMAG magnetometer network. The SuperMAG magnetometer network in the northern hemisphere (up to 100 stations) improves the traditional auroral electrojet (AL) network (12 stations) (Newell and Gjerloev, 2011a).

Newell and Gjerloev (2011b) distinguish recurrent and isolated substorms. While recurrent substorms appear in groups with less than 82 min between their onsets, the isolated substorm onsets are separated by at least 3 h. Only the isolated substorms are used in our investigation, as this helps to avoid two or more substorms overlapping each other. Contrasting the isolated

substorm periods we also use time periods without any substorms (no-substorm period). The total number of substorm onsets for our period constitutes 15 316 events. Defining 30 min after an onset as typical length of a substorm, we end up with 10.4% of the whole period being generally substorm-influenced (while the rest is no-substorm). However, just 1.87% of the whole period can be attributed to isolated substorms.

It should be noted that with this technique we are not able to separate different substorm phases nor can we distinguish different types of substorms. Independent from substorm phase, the proton aurora is displaced equatorward of the electron aurora for dusk local times, and it is poleward for dawn local times. In the onset region however, proton and electron precipitation depends on the substorm phase and may even be colocated (Mende et al., 2003). Thus the results represent a mean substorm value.

### 2.3.1   Kp-binning of particle data

The Kp-index is a three-hourly index estimating the geomagnetic activity (Bartels et al., 1939). In contrast to the AL/SML index which describes the auroral electrojet activity, the Kp-index is sensitive to several current systems (e.g. the ring current) and thus describes the magnetospheric activity with a more global perspective.

The particle data has been binned into 11 partly overlapping Kp-level groups: 0–0.7, 0–1, 1–1.7, 1.3–2.3, 2–2.7, 3–3.7, 4–4.7, 5–5.7, 6–6.7, 6–9 and 7–9. As the Kp-levels are not equally populated (low Kp-levels occur more frequently than e.g. 6–6.7), the amount of satellites is not constant, the substorms are not evenly distributed in time and the local time sectors are not evenly covered, single data points (with 1 h MLT-resolution, 2 degrees latitudinal resolution and the Kp-binning) may contain a different amount of the 16 s averages.

## 3   Particle flux map

Binning of particle flux strongly depends on the coordinate system. Some features are determined by the inner magnetic field and thus co-rotating with Earth, while others are influenced by the interaction with the solar wind and according to that fixed in relation to the Sun and to the (magnetic) local time. Since we will use the modified APEX coordinates in this paper we will have a look how the particle flux representation differs to geographic coordinates and which aspects can be best described in the two systems.

Figure 1 shows the TED proton band 11 in geographic coordinates (top row) and modified APEX 110 km coordinates (bottom row). The left column shows latitude against longitude while the right column shows latitude against MLT. No selection according to Kp-level or substorm intensity has been made, while all available data from METOP 2 and POES 15, 16, 17 and 18 for the years 2001–2008 has been included. This allows a spatial resolution of 3.75 degrees longitude (or 15 min MLT). Please note that latter figures show a reduced longitudinal resolution of 15 degrees (or 1 h MLT) only to avoid statistical noise in e.g isolated substorms periods.

Most obvious in the geographic representation (Fig 1, top, left) is the South Atlantic Anomaly (SAA, located roughly between 280 and 360 degrees East and -45 to 0 degrees North). Being a dip in the geomagnetic field, the SAA allows energetic

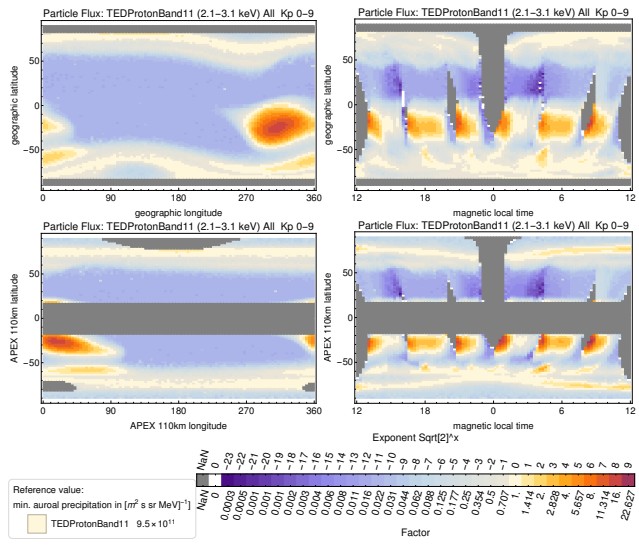

**Figure 1.** Particle flux in the TED proton band 11 in geographic and modified APEX 110 km coordinates. The color scale marks the minimum flux in the auroral oval in beige. The neighbouring color indicates that the flux is a factor $\sqrt{2}$ apart (the neighbour after that a factor 2).

particles in the radiation belt to reach altitudes low enough to be reached by the satellite's orbit. According to Rodger et al. (2010, Fig. 1) the particle population in the SAA consists of particles precipitating in bounce and drift loss cone as well as trapped particles. Thus the high flux values are not necessarily connected to high particle precipitation. As the SAA is a geomagnetic feature it is co-rotating and thus best represented in the geographic coordinates, while a MLT based coordinate

5 system intermixes SAA patterns with non SAA patterns on the same latitude. However, in contrast to our expectations the SAA in MLT representation is not evenly smeared out over all latitudes. A detailed discussion on this follows in Section 3.1.

A feature that is connected to the SAA is the particle precipitation of the drift loss cone. Particles drift around Earth and bounce between the mirror points. These mirror points get to lowest altitudes where the magnetic field is weakest. Since the geomagnetic field around the SAA is weak the dominating particle precipitation zone is where these field lines have their

10 foot points. In Fig. 1 (top, left) this can be clearly identified South-East of the SAA. In the northern hemisphere the particle precipitation due to the drift loss cone is less dominant, but still visible.

Figure 1 (top, right) shows the same data on a geographic latitude vs. MLT grid. As the auroral oval is not visible as an oval any more but mixes up local time differences and the latitudinal variations that can already be seen in Fig. 1 (top, left).

Apart from that the geographic representation is not very helpful. Due to the satellites' inclination the poles are not covered

15 and typical pattern as the auroral precipitation is meandering.

Switching to magnetic modified APEX 110 km coordinates (see Fig 1, bottom, left) straightens the auroral oval and mostly removes the longitudinal dependence except for the SAA and the drift loss cone in the South of the SAA. Consequently we can replace the APEX longitude by MLT (see Fig. 1, bottom, right). Features that depend on MLT now become visible and the auroral oval itself does not show a hemispheric dependence. The SAA and the drift loss cone, however, are now smeared out

and still produce a hemispheric asymmetry. The drift loss cone that is located at a distinct modified APEX 110 km longitudinal range even appears as an double auroral structure at the same latitude but covering all longitudes. Which of course is an artifact of this kind of MLT binning.

Most obvious in the modified APEX/MLT coordinates (see Fig. 1, bottom, right) are the local time dependencies in the auroral zone as well as at lower latitudes. Substorm dependent precipitation that mostly appears during night time can also be identified (see following sections).

Some regions in modified APEX/MLT coordinates will never be reached as the local time coverage is limited. This holds for the midnight hours in the northern hemisphere as well as for noon at low latitudinal southern hemisphere. The equatorial region seems not to be covered, however, is not a data gap. The flux is mapped to the latitude where the guiding field-line hits 110 km. Since the satellites cross the (dip) equator at 850 km all field lines peak below that point are not covered (<19 degrees N/S). Since the magnetic poles are shifted to the geographic ones the satellites' inclination does not limit the polar coverage.

As a consequence of the regional coverage and SAA influence we will select the Southern hemisphere auroral zone for further investigation.

### 3.1 Why is the SAA not evenly smeared out over all longitudes?

If we would take a look at the footpoints of a solar-synchronous satellite in local time we would see that it always crosses a particular latitude e.g. the equator at one particular local time in ascending mode (and another, at the equator 12 hours later, in descending mode). At high latitudes it crosses 12 local time zones on a few latitudes, but still, the next orbit will exactly match the first (except if the orbit moves, which also happens to the POES/METOP satellites, but on longer time scales). Looking at the footpoints of the same satellite in MLT changes quite a bit. Given that the MLT zones are based on magnetic longitude and the magnetic poles being shifted, it means that the MLT-footpoints especially in high latitudes differ significantly from one orbit to the next. Due to the POES inclination of 98.5 degrees the satellite may at maximum reach the northern magnetic pole. The southern magnetic pole however, may not only be reached but even passed.

Thus there are two options how the MLT during an orbit may develop in the southern hemisphere: If the satellite's longitude is far from the magnetic pole the orbit will not pass the magnetic pole and the MLT will gradually increase by 12 hours till it reaches the equator in ascending mode again. Let us call this "ascending MLT". In the other case ("descending MLT"), the southern magnetic pole will be passed and the MLT zones will be flown through in the opposite direction, decreasing MLT by 12 hours till reaching the equator in ascending mode again. Since the southern magnetic pole is somewhat south of Australia a significant fraction of the orbits passing it will cross the SAA in descending mode (but not in ascending mode). The opposite is true for the ascending MLT path, which includes a significant fraction of orbits that pass the SAA in ascending mode.

In case multiple satellites are used this does not affect high latitudes, but in low latitudes the situation is different. Since the satellites are crossing the equator at two specific local times (for ascending and descending mode, being just slightly broader in MLT), MLT coverage at the equator is limited to these points. They however may be reached in ascending mode (or left in descending mode) by ascending or descending MLT paths. In Fig. 1, top right (or bottom right) the equator is crossed at six

different smeared out MLTs. While the ones on the left (13, 17 -two satellites- and 21 MLT) represent the descending mode, the ones on the right (1, 5 and 9 MLT) are in ascending mode.

The ascending MLT path now connects e.g. the low flux right edge of the 21 MLT equatorial crossing with the high flux (SAA) left edge of the 9 MLT equatorial crossing. The descending MLT path on the other side connects e.g. the high flux
(SAA) left edge of the 21 MLT equatorial crossing with the low flux right edge of the 9 MLT equatorial crossing.

In sum the ascending and descending MLT paths cause the left edge of an equatorial crossing to be affected by the SAA, while the right edge is not. Any MLT analysis of latitudes that show longitudinal variations will suffer from the fact that longitudes contribute very unevenly to the MLT zones in the polar orbit of POES/METOP. Given that the SAA is the dominant flux source in low latitudes this hampers a MLT flux analysis here. Effects may also be seen in the drift loss cone, where longitudinal flux
variations are expected. In high latitudes however, just minor longitudinal variations (in magnetic coordinates) are expected (see Fig. 1, bottom-left, auroral zone). Consequently it just has minor affects on the results, but not the overall findings and trends. Additionally this effect gets counterbalanced by broader MLT coverage and multiple satellites in high latitudes.

## 3.2  Determination of the auroral oval

In some parts of the paper we will refer to APEX 110 km latitude or MLT locations of the auroral oval or its flux maximum
and minimum. These locations have been determined automatically. A routine determines the maximum flux for each MLT-bin within the typical auroral latitude range. This results in a preliminary auroral oval. Then the latitudinal differences between MLT-predecessor and successor are determined and in case of large outliers a point is assumed to be a spike in the data and replaced by the next biggest flux-bin in that MLT zone. In case that more than 7 points have to be replaced for a auroral oval the according channel-Kp set is neglected. In sum this ends up in a well-working detection algorithm for the auroral oval
and allows us to find its minimum and maximum flux, or their ratio. A sample is given in Figure 2. All locations have been cross-checked manually.

## 4  Particle flux at high latitudes

Figure 3 shows the precipitating electron flux at high latitudes in the southern hemisphere. The southern hemisphere has been chosen to avoid the data gaps between 23 and 1 MLT in the northern hemisphere (see the Fig. 1, bottom right). Apart from that,
northern and southern hemisphere do not show significant differences in APEX coordinates.

The color scale is logarithmic with a base of two, meaning the threshold to the adjacent color is a factor of 2 apart. The reference value has been set individually for every channel to the lowest occurring value inside the auroral oval. Thus local time differences can be easily identified and quantified. No-substorm periods (left panel) and isolated substorm periods (right panel) for all electron channels are given here.
Figure 4 shows the same as Figure 3 but for protons.

Comparing the two panels of Fig. 3 and 4, we can identify:

1. Typical pattern in low energetic channels (see Section 4.1),

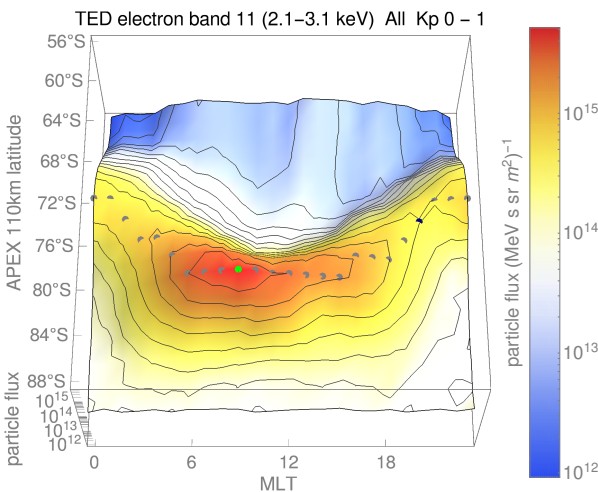

**Figure 2.** Sample for an auroral oval fit. The gray dots represent the position of the auroral oval. The green (9 MLT) and black (20 MLT) dots indicate maximum and minimum of the auroral oval, respectively.

2. Typical pattern in high energetic channels (see Section 4.2),

3. Kp-dependence of the auroral MLT-asymmetry (see Section 4.3),

4. Auroral oval asymmetry during substorms (see Section 4.4), and a

5. Latitudinal displacement of the maximum auroral flux depending on Kp and energy (see Section 4.5).

## 4.1 Typical pattern in low energetic channels

Low energetic proton and electron channels, namely TED electron band 4 and 8 as well as proton band 4, 8 and 11 show a very different spatial pattern than the higher channels.

The maximum flux in the auroral oval appears in the day sector. TED electron band 4 and 8 peak between 6 and 17 MLT. This agrees e.g. very well with the monoenergetic electron number flux for low solar wind driving (Fig. 7 in Newell et al., 2009).

The proton bands are even more concentrated around noon but show an additional slight increase from noon via the morning sector towards midnight. Since this is completely opposite to the higher channels we will have a look at the source region.

The main precipitation of low energetic electrons (<1 keV) at daytime (e.g. 76–80 S, 6–13 MLT for TED electron band 4) most likely origins from the poleward edge of the cusp, referring to Sandholt et al. (1996); Øieroset et al. (1997); Sandholt et al. (2000) who attribute this as source region during periods with northward IMF (which in our study mostly refers to no-substorm periods as southward IMF triggers substorms).

In contrast, during periods with isolated substorms the particle flux is shifted by 2 degrees to the equator. The source in this case is the equatorward edge of the cusp which has been identified as corresponding source region in periods of southward

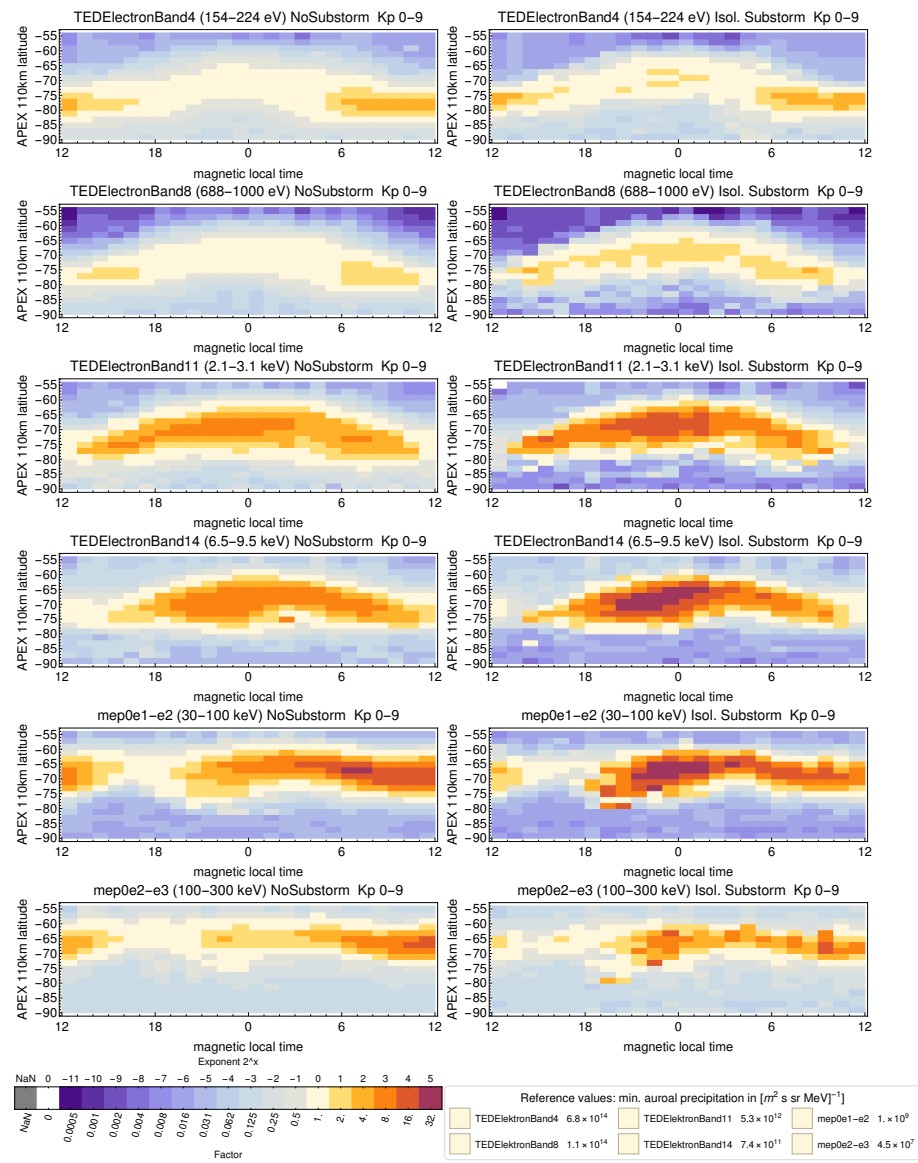

**Figure 3.** Electron flux in various channels at high latitudes on the southern hemisphere. Left panel shows periods without substorms, right panel gives periods with isolated substorms only.

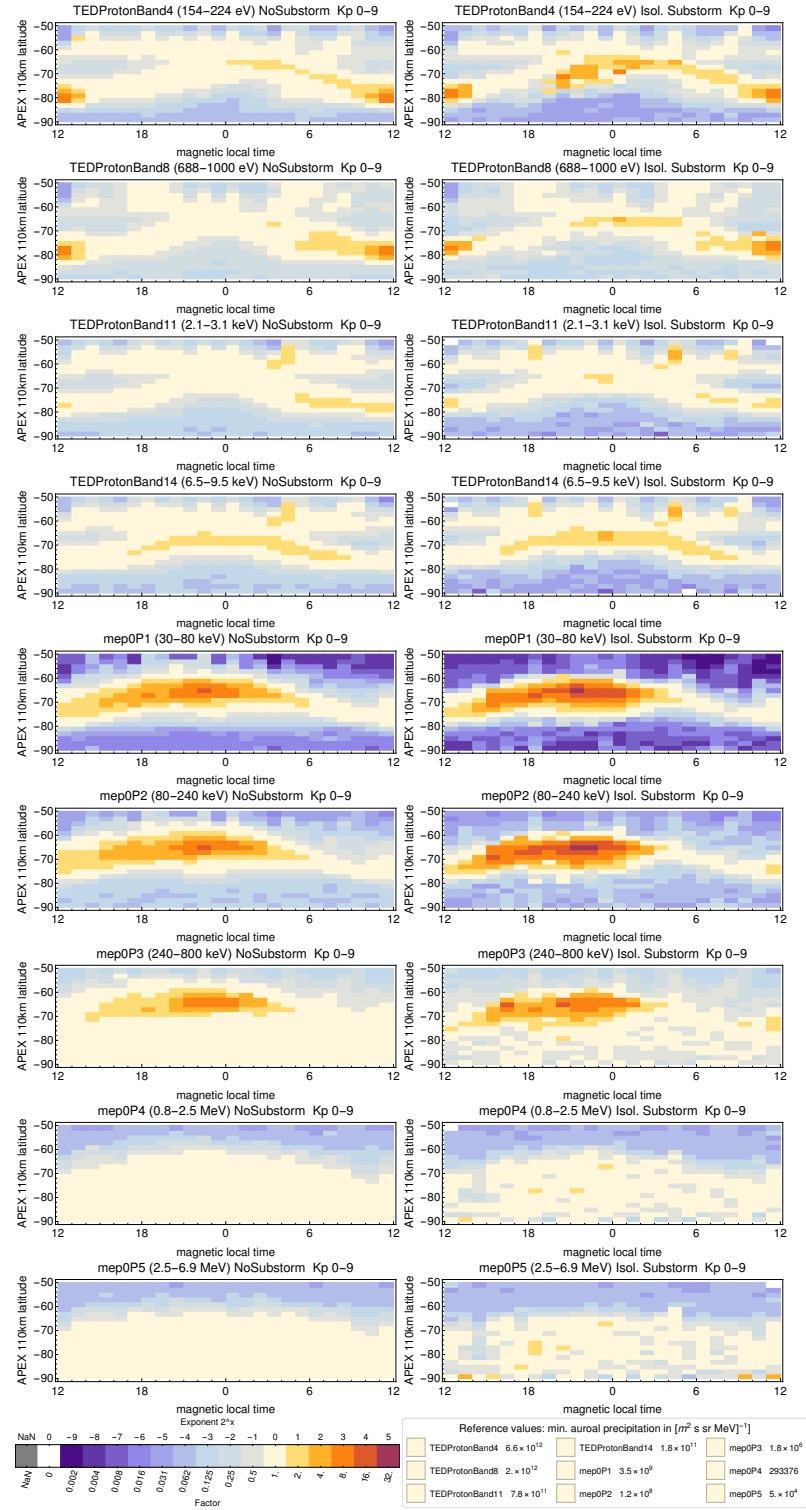

**Figure 4.** Proton flux in various channels at high latitudes on the southern hemisphere.

IMF by Sandholt and Newell (1992); Sandholt et al. (1998). A sketch including the source regions may be found in Newell and Meng (1992, Fig. 2), even though the regions are labled with Mantle and Cusp here.

In sum our findings support that high numbers of low energetic particles enter the magnetosphere preferentially on the front side through cusp and other boundary layers (Newell et al., 2009).

Additionally low energetic protons (TED proton band 4–14) show a second oval structure (approx. 50–65 S), which is associated with the drift loss cone (see the Section 3) and thus geographically localized near the SAA. The second oval structure itself is a artifact of the MLT-binning (see Fig. 1).

## 4.2   Typical pattern in high energetic channels

The high electron channels (above >2 keV) show a displacement of the main particle flux with MLT from midnight (TED
electron band 11) via morning sector (mep0e1-e2) to day sector (mep0e2-e3).

Concerning protons, between TED proton band 14 and the following channels (mep0P1 to mep0P3) the main particle flux shifts from midnight to the evening sector, which is oppositely directed to the electron displacement.

A potential explanation for the displacement in the higher electron channels (and the opposite directed shift of the protons) is the westward partial ring current in the night side which is closed by field-aligned currents (Birkeland Region 2) into the
ionosphere (Lockwood, 2013; Milan et al., 2017). While electrons in ring current drift eastwards and thus may precipitate predominantly in the morning sector, the protons undergo a westward drift and mainly precipitate in the evening sector. The energy dependence might be due to different drift velocities (Allison et al., 2017). A drift of electron precipitation (>20 keV) towards the dayside has also been reported by Matthews et al. (1988); Newell and Meng (1992); Østgaard et al. (1999) and is associated with central plasmasheet electron injections in the midnight region.

The resulting auroral asymmetry also depends on Kp-level, as shown in Section 4.3.

## 4.3   Kp-dependence of the auroral MLT-asymmetry

Even without Kp-dependence Figures 3 and 4 reveal a channel (energy) dependent MLT-asymmetry of the particle flux and that the range of this dependence changes with particle energy.

While e.g. the two lowest TED electron channels (band 4 and 8) show just minor MLT variations, it varies by more than one
order of magnitude in the higher particle energies.

The proton flux shows distinct MLT dependence, ranging from just minor variations in TED proton band 11 and 14 (as well as practically no MLT variation in the highest proton MEPED channels) up to about one order of magnitude in the lowest and medium particle energies (TED proton band 4 and 8, MEPED mep0P1 and mep0P3).

During isolated substorms the maximum local time differences are similar or a factor of 2 higher.

However, we noticed that the MLT-asymmetry is not constant over different Kp-levels. This section will emphasis on the impact of Kp-levels using Figure 5, which is based on the auroral oval determination algorithm from Section 3.2 and presents the ratio between maximum and minimum auroral oval flux (or in other words the asymmetry of the oval) depending on Kp-

level for every channel separately. Actually the channels have been grouped by their Kp dependency. All these findings are based on the whole period disregarding substorm or not.

The upper panel shows the 2 lowest electron channels and the lowest proton channel which all have a declining flux asymmetry with increasing Kp. The 6–6.7 Kp-bin is enhanced here, but we should keep in mind that this levels are occurring rarely and may suffer from bad statistics. A reason for the decline might be that the cusp inflow is not increasing with Kp as the rest of the auroral flux. Thus its relative fraction declines and subsequent declines the asymmetry.

The middle panel shows all particle channels that have an increasing flux asymmetry with Kp, as they are: all remaining electron channels and the proton channels TED band 11 and mep0P1. Given that high geomagnetic disturbance should be linked with enhanced acceleration, scattering and substorm processes increasing asymmetry in the affected regions suggests itself.

The lowest panel gives the flux asymmetry dependencies of the remaining proton channels that are less distinct. It seems that there is a domain change at about 3.3 Kp since the asymmetry of TED proton band 14 and mep0P2 has a negative correlation below 3.3 and a positive one above. For the channels TED proton band 8 and mep0P3 the relationship is opposite.

The two highest energy channels (MEPED mep0P4 and mep0P5) do not show MLT variations as seen in Figure 4. Particle precipitation is limited to solar proton events. Since these particles enter the ionosphere via open field lines there is no latitudinal focussing but a homogeneous precipitation within the polar cap. Which is the reason why the auroral oval fit-routine failed and these channels are not listed in Fig. 5.

In sum, the maximum MLT-asymmetry depends on Kp:

– for very low energy (proton and electron) it decreases with Kp,

– for higher electron channels it increases with Kp,

– for higher proton channels the Kp-dependence in ambiguous, but in general the asymmetry is significantly smaller than in the electron channels.

## 4.4 Auroral oval asymmetry during substorms

This Section discusses the changes during substorm periods based on Figures 3, 4 and later on Fig. 6.

In general, the particle flux during isolated substorms is similar to no-substorm periods, but superimposed with substorm-specific night-side (20–2 MLT) particle precipitation which reflects the substorm electrojet manifestations (Lockwood, 2013; Milan et al., 2017) (see Figures 3 and 4). In terms of particle acceleration this is the same region that shows Alfvén waves (compare Fig. 4 in Newell et al., 2009) or Alfvénic acceleration (Fig. 14 in Dombeck et al., 2018).

The electron and proton flux intensity at midnight sector outnumbers the no-substorm values at the same place by factor 2 to 4. For mep0P1 to mep0P3 also the evening sector is slightly increased during substorms. Given the flux increase in the midnight sector the maximum auroral flux during a substorm can mostly be seen around 0 MLT (see Figures 3 and 4).

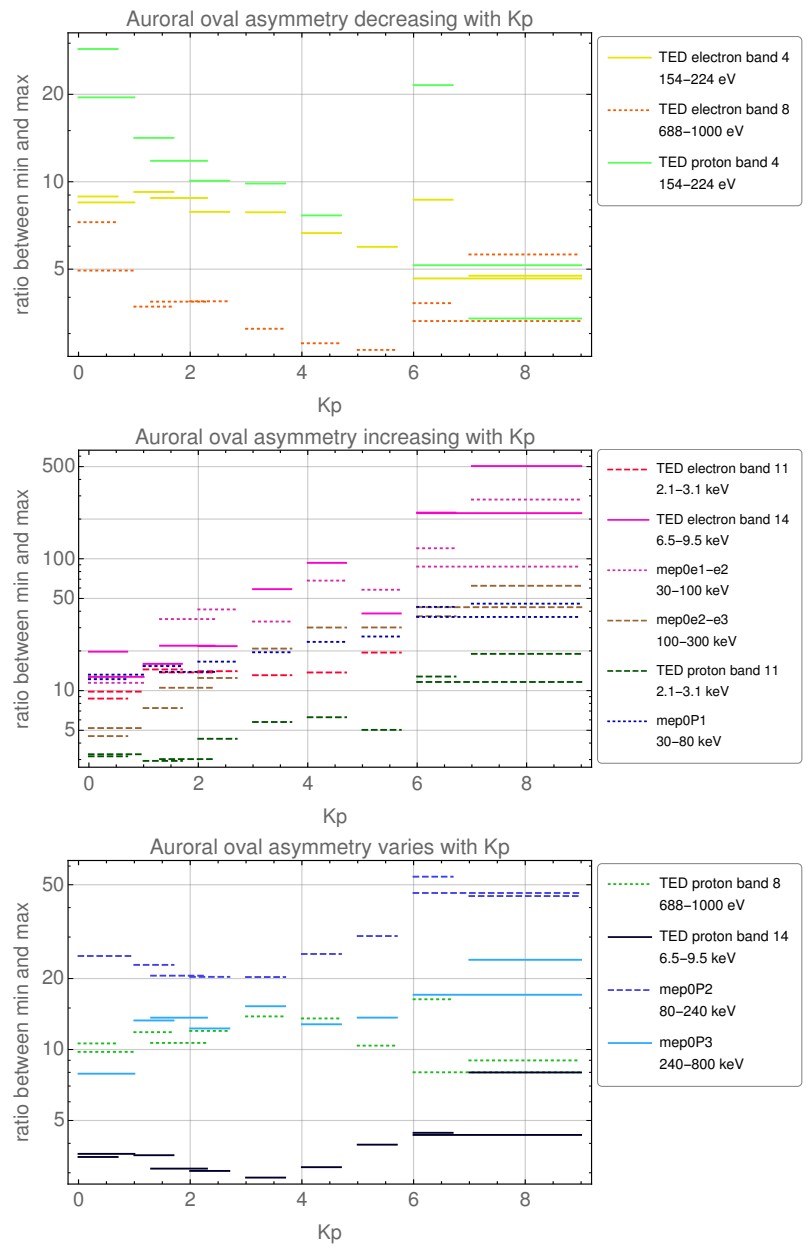

**Figure 5.** Dependence of the auroral oval asymmetry with Kp. Top: Channels which auroral oval flux shows a negative correlation with Kp, middle: positive correlation, bottom: no clear correlation.

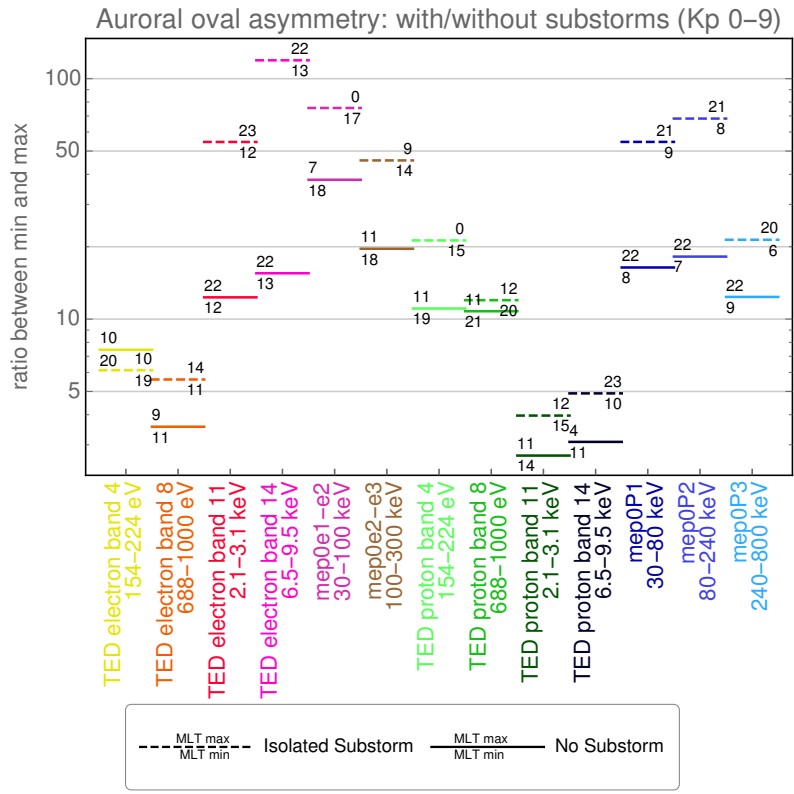

**Figure 6.** The auroral asymmetry is shown as reatio between maximum and minimum auroral oval flux. The number above a specific ratio states the MLT where the maximum is detected while the number below that ratio indicates the MLT of the minimum.

Additionally noon sector electron fluxes decrease during a substorm, which is clearly seen in all upper channels (from TED electron band 11 to mep0e2-e3). The noon sector flux decreases most probably because dayside particle precipitation occurs often during northward orientated IMF which is not usual for substorms (see Figure 3).

In contrast to the electron fluxes the day-sector proton fluxes do not significantly depend on substorm activity (see Figures 4).

Figure 6 presents how the asymmetry depends on substorms. Since an 8 year period does not contain enough values for substorms in rare Kp-levels we neglected the Kp-level here and compared isolated substorm to no-substorm periods.

Except for TED electron band 4 (where there is no significant difference between substorm and no-substorm periods) all other channels have an increased auroral oval asymmetry during isolated substorms. The numbers above and below the marked flux ratio indicate the MLT location of the minimum (below) and maximum (above). We can identify that the maximum flux during a substorm shifts to the midnight sector (if not already there in no-substorm periods) e.g. for mep0e1-e2, TED proton band 4 and 14.

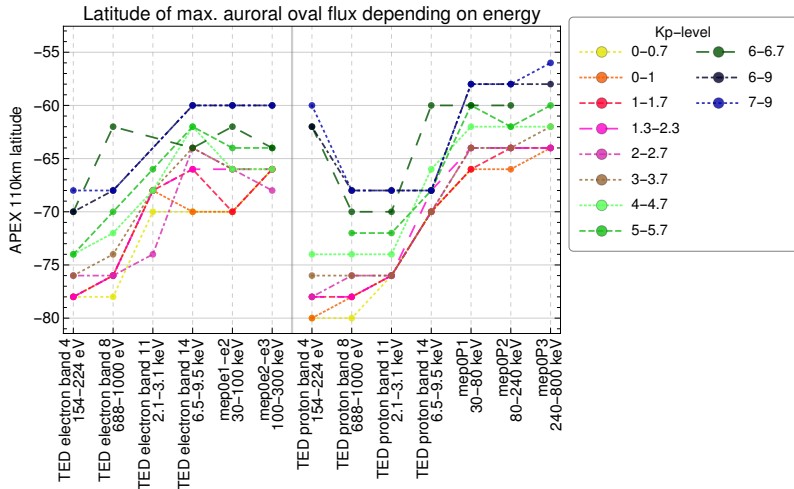

**Figure 7.** The modified APEX 110 km latitude of the maximum flux in the auroral oval is shown. Colors indicate specific Kp-levels. Left hand displays the energy dependence of electrons, right hand that one of the protons.

For TED electron band 4 and 8 (as well as TED proton band 8 and 11) the substorm enhancement is also seen in the night sector, but it does not overshoot the dayside flux (see Figures 3 and 4), while the substorm enhancement in the night-sector of mep0e2-e3 is in the same order as the 9–12 MLT flux.

This agrees with Newell et al. (2009) stating that the low energetic particles that enter the magnetosphere in the day-side are 5 accelerated by the magnetotail and precipitating at the night-side during substorms.

The asymmetry in both, the electron and the proton spectra (as well as during no-substorm or substorm periods) shows a local minimum in middle TED channels (TED electron band 8 and proton band 11) as well as a local maximum at higher energies (TED electron band 14 or mep0e1-e2 for electron and mep0P1 or mep0P2 for protons). At even higher energies the asymmetry declines again.

10 **4.5  Latitudinal displacement of the maximum auroral flux depending on Kp and energy**

Figure 7 presents how the latitude of the maximum auroral oval flux varies with particle energy and Kp. In advance we should note that Fig. 7 may not be over-interpreted since latitudinal displacement of the maximum flux is mainly an effect of a MLT-shift (see Fig. 3) that causes the strong latitudinal change between TED electron band 8 and TED electron band 14 for electrons and TED proton band 11 and mep0P1 for protons. Thus the main precipitation zone undergoes a strong latitudinal change, but 15 it does not necessarily describe a latitudinal change in the auroral oval.

The figure has been derived by the auroral oval determination method discussed in Section 3.2 and displays the latitude of maximum auroral oval flux.

Except for some outliers, most of them belonging to TED proton band 4 during high Kp levels ($> 6$), the graphs show a clear equatorial dislocation with increasing energy. The 110 km APEX latitudinal range at a specific Kp-level is about $10°$ for

electrons and 12-16° for protons. This dislocation however appears to be stepwise: TED electron band 4 and 8 are almost on the same latitude as well as TED electron band 14, mep0e1-e2 and mep0e2-e3 share the same latitude. For protons TED proton band 4, 8 and 11 are almost on one latitude and the higher particle energies mep0P1 and mep0P2 are co-located. This implies that these particles origin from the same souce population.

However, there is a noticable latitudinal shift with particle energy even for the particle channels that appear co-located. For 8 out of 11 Kp-levels there is an equatorial shift of 2° or more between TED electron band 4 and 8. For protons a latitudinal shift can be recognizend between TED proton band 4 and 11 or between mep0P1 and mep0P3.

Every color graph represents the spectral location of the maximum flux latitude for a certain Kp-range. Thus we can infer that increased geomagnetic disturbance (high Kp values) causes a dislocation of up to about 8° towards the equator.

Concerning the outliers in TED proton band 4, for low Kp-values there is a clear flux maximum at noon, which is located at rather high latitudes (compare Fig. 3). At high Kp, the MLT asymmetry declines and then flips. Consequently the maximum flux for high Kp-levels is not in the day sector and thus at significantly lower latitudes, but even between the outliers and the maximum flux location in mep0P3 (which is in a similar MLT-region) we can recognize a latitudinal shift.

In sum, there is an equatorial shift of the main precipitation zone with increasing Kp and increasing particle energy, while
the latter is primarily due to a shift in MLT and only secondarily due to a latitudinal shift of the auroral oval itself.

## 5   Summary

In this paper we presented the MLT-distribution of energetic particle flux/precipitation into the ionosphere in combination with different substorm activity.

We could identify low energetic particles to predominantly precipitate around local noon, supporting the idea that they enter
the magnetosphere through the cusp. During substorms the maximum particle flux is shifted by 2 degrees to the equator.

Higher particles energies show a different behavior. Electrons (>2 keV) mainly precipitate at midnight but with increasing particle energy the maximum flux shifts via morning-sector to the day-sector. Maximum protons flux on the other hand, shift from the midnight-sector to the evening-sector with increasing energy. A drift of electron precipitation (>20 keV) towards the dayside is associated with central plasmasheet electron injections in the midnight region.

There is an energy dependent auroral asymmetry. While the low energetic electrons have just a minor asymmetry it enhances to more than one order of magnitude for the higher electron channels. For low energetic protons the cusp precipitation causes an asymmetry of about an order of magnitude. Above that energy the asymmetry first declines (to factor 2 in TED proton band 11 and 14) and than enlarges again with the MEPED channels 1 to 3 (more than an order of magnitude). For highest proton channels the asymmetry disappears as these particles are not linked with auroral precipitation. During substorms the maximum
flux is similar or a factor 2 higher.

The auroral asymmetry is Kp-dependent. For low energetic particles the asymmety declines with Kp probably due to lack of cusp precipitation during high Kp-values, while it increases especially (stronger) for higher electron channels probably due

to increased acceleration and scattering processes. For medium and high energetic protons the development of the asymmetry with Kp is not that distinct, there might be multiple processes involved.

During substorms the no-substorm flux seems to be generally superimposed by substorm-specific night-side particle flux. However, the noon-sector fluxes depend on particle species. For protons they seem to be independed from substorm activity, while for electrons they decrease during a substorm.

Also we noticed a Kp and energy dependent equatorial shift of the main flux latitude.

*Competing interests.*   The authors declare that they have no conflict of interest.

*Acknowledgements.*   The authors acknowledge the NOAA National Centers for Environmental Information (https://ngdc.noaa.gov/stp/satellite/ poes/dataaccess.html) for the POES and METOP particle data used in this study and give many thanks to the SuperMAG team (http: //supermag.jhuapl.edu/) and their collaborators (http://supermag.jhuapl.edu/info/?page=acknowledgement).

The work is supported by the DFG project WI4417/2-1.

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
