# Peer review of "Magnetic local time asymmetries in precipitating electron and proton populations with and without substorm activity"

_Annales Geophysicae, 2019_

## Referee Comment (RC1) · Anonymous Referee #1 · 28 May 2019

The paper uses POES and METOP data to evaluate how the precipitation of protons and electrons varies with magnetic local time during both isolated substorms and in the absence of substorm activity. Data collected during the declining phase of cycle 23 has been used, and measurements from multiple energy channels evaluated. Overall this is an interesting piece of work and generally well written. The introduction and motivation in particular is very well thought out.

My major concern for the paper is that one of the authors main results, the Kp (or lack of) dependence of proton and electron precipitation, is discussed in the abstract as well as the summary section, but any figure actually showing this has been omitted

from the paper. While an indication is given in table 2, this is difficult to assimilate and results considering different Kp bins need to be presented as a figure for this result to be claimed.

All other comments I have on the paper are more minor.

Line 19, section 1, discusses how the SML index was used to define the substorm onsets. While the reviewer agrees that SML is a good choice to define substorm onsets, perhaps the authors could elaborate on why SML was used instead of the AE index as in the Reeves et al., 2003 study.

In section 3, line 11-12 reads "the SAA allows energetic particles in the radiation belt to reach altitudes low enough to be reached by the satellites orbit". Considering that the authors are solely using the T0 flux measured by POES and METOP, Figure 1 in Rodger et al., 2010 (Use of POES SEM‐2 observations to examine radiation belt dynamics and energetic electron precipitation into the atmosphere, JGR, doi:10.1029/2008JA014023) would suggest that, even over the SAA, the T0 measured flux is still precipitating.

In Figures 2, 3, and 4, as well as in the text, could the different energy channels be referred to by the energy range covered rather than the channel name? This would make the results easier to interpret without constantly flicking back to table 1.

When discussing figure 2 in section 4, point c and d mention that the noon sector flux decreases during a substorm. Could the authors speculate on why this is?

When discussing Figure 3, the first point claims that protons also show an equatorial movement of the main precipitation zone with increasing particle energy. More direction to this in the figure is required here for the reader as I do not see this.

The second point made when discussing Figure 3 was that there is a second auroral oval. This is then stated to be an artefact of the MLT binning. Could the authors add further explanation to justify this here?

[Figure]

In the text, table 3 is not discussed or explained. Perhaps it is not needed? Otherwise the relevance needs to be discussed.

Additionally, there are a number of typographical errors in the manuscript: Page 4, line 3: 'furtunately', line 15: 'trapped particles in low altitudes' should be 'at low altitudes'. Line 25: 'mainly located in about 110 km altitude' should be 'at about 110 km...'. Line 34: you have not defined the QD acronym - please write out quasi-dipole. Spelling of 'avoid' throughout the manuscript is wrong. Page 5, line 20: 'independend'. Page 6, line 29: 'Substorm depended precipitation...'

---

## Referee Comment (RC2) · Anonymous Referee #2 · 25 Jun 2019

The paper describes electron and proton differential number flux 1/(m^2-s-sr-MeV – according to the figure labels) measured in various energy channels by detectors on POES and METOP from 2001-2008 in the radial direction away from Earth for electrons and 9 degrees from that in protons. It presents the results of these by field line mapped latitude at 110 km altitude and by magnetic local time (MLT) for non-sub-storm times and isolated sub-storms. It also makes mention of comparisons to geomagnetic activity (Kp), but does not present any data in this regard.

The data in the paper could have potential contributions to the field with additional work and resubmission.

The data itself is a contribution to the field providing the same type of measurements and analysis over a wide energy range for both electrons and protons. However, the difference in scales for the different channels and the substantial difference in energy widths of the channels makes inter-comparison between the channels difficult other than relative locations of peaks and troughs and differentials, which are the focus of the paper results. Multiplying by the geometric mean of the channel energy ranges and dividing by the channel energy width would give a reasonable proxy for energy flux, which could be put on the same scale and would allow direct comparison of levels between the energy channels, which would add to the contribution from the presentation of the data.

The data would be a contribution (and could be improved as indicated), but a good amount of additional is required to other facets of the paper.

In particular, there are two major components that need to be significantly improved. One is the need to put the data in appropriate context of previous work, empirical and physical theories. The results also would need to be appropriately put into context. The second is a loose use of language in important ways that causes misrepresentation of the data and focus of the paper.

In particular, the has been a substantial amount of work done of the latitude and MLT distributions of electron (and proton) precipitation, particularly at auroral latitudes and at the lower (TED equivalent) energies, such as Cattell et al, 2006, Newell et al, 2009, and more recently, Dombeck et al, 2018 and the references within these. There is also a substantial number of papers on the MLT and latitude dependence and relationship between Alfvén waves and electron precipitation, by the likes of Keiling, Chaston and Hatch. Although these later (Alfvén wave papers) only relate to one small energy band and one part of the precipitation in the channel, so comparison to the presented results would be indirect, so perhaps could not be included in detail. However, the Cattell, Newell and Dombeck papers all show latitude versus MLT plots of precipitation that can and should be compared to the data and results presented in this paper.

To get precipitation (other than drift lose cone affects in the SAA, which the authors are pointedly not trying to investigate). acceleration and/or wave scattering into the loss cone is necessary. These mechanisms have been discussed and investigated in the literature for decades, and they are dramatically different for the different species and energy channels discussed. All of this has direct relevance to the presented data and the proper and useful interpretation of the results from those data, and the data and results should be discussed from firmly within that context.

For the second major issue, the loose use of language, the most egregious issue is the use of the loose use of the word precipitation. In particular, the title and abstract indicates that the paper is about precipitation, but even in the best case this is only half of the presented results, the highest latitude results. As the authors state, the lower latitudes are primarily measuring trapped populations. Therefore the title does not properly describe the paper. Even at the higher latitudes, what is being measured is the downgoing population (centered at various pitch angles depending on latitude, population, and relative potion of the spacecraft to Earth's magnetic pole). While this is still mostly "downward", downward does not necessarily mean precipitating, as the authors point out in relation to their discussion of the SAA. The authors need to be clear what it is that the data presented are. The title and discussion should be adjusted accordingly.

Additional loose uses of language include discussions of "source" particles, and "qualitative" and "quantitative" results. In particular, there are really only two sources of particles that are being detected: the solar wind/sun, and the ionosphere/Earth's atmosphere. One could argue that populations that are trapped in the ring current, plasmasphere, radiation belts, plasmasheet, etc, are different pools of particles that are "sources" for the particles measured in study. However, that argument has not been made in the paper. The authors also appear to using the word "source" to allude to both populations pool that the measured particles came from/belong to, and for the mechanisms that cause them to precipitate/be observed in the data. This really needs

to be clearly described in the paper.

The authors also make a pointed distinction between "quantitative" and "qualitative" results in the paper. In nearly all cases, however, the results described in the paper are qualitative descriptions of the quantitative presented data. The only except to this is the minimum to maximum differences by MLT, which is purely quantitative. Regardless, the use of "qualitative" and "quantitative" as descriptors for the results is unnecessary and as used highly confusing (and inaccurate). As such they should likely just be removed.

At a very minimum these two major issues need to be addressed: the data and results need to be presented within the proper context (a useful review of the TED energy range precipitation is in by Frey, 2007), and a more precise description of what is presented/covered in the paper is required.

Until the paper is reworked into the proper context, it is difficult to determine whether the conclusions are significant (currently they are not), how important the contribution is, or if the length of the paper is adequate. The language is mostly fluent, but does need to be made more precise in ways, as mentioned.

The figures are of adequate size, although the text in them is too small in some cases. The presentation is clear and organized, although missing the context for appropriate organization.

Several other things that should be addressed in the next version of the paper, include the following:

A figure showing coverage by MLT and latitude would be very helpful.

A thorough discussion of the minimum count levels of the instruments, and how the noise associated with 1 count levels, interpretation of zero counts, and differences between instruments and channels with regard to this are addressed should be included.

The field of view of the detectors should be discussed as well as the effects this has in interpretation of the results, from both the perspective of mirroring particles as well as

potential anisotropic distributions.

The word "moves" is used to described the differences between energy channel results. This is not the appropriate word, they are different populations, the features are not "moving" in any sense.

The "precipitation zone" is not at lower latitude with higher energy in general as indicated in the paper. The peak latitude appears identical for TED 8, 11, and 14, for example. The peak "precipitation zone" latitude is related to where particles of these energies reside (radiation belts, etc.) and/or where the acceleration mechanisms that cause those energies occur. This will be clarified once the word is put into appropriate context. It is unclear that there is a physical meaning to the particle energy to latitude of peak flux, so perhaps this relation should be omitted. If it is included, it should be demonstrated with a statistical plot of energy versus peak flux latitude, or some such, from the data.

The plume has no relevance to the discussion of the 9-10 MLT hotspot, and should not be included in that discussion. This hotspot has been observed before in the Cattell, Newell and Dombeck et al, papers, for example, as well as tangentially addressed in Frey, and should be discussed within the context.

Other details that should be addressed include:

Page 1, Line 8: The sentence ending with "how" should be reworded.

Page 1, Line 13: "main link" should be replaces with something like "a primary link". Solar UV input has much more affect on atmospheric chemistry than particle precipitation, and even affects the precipitation.

Page 1, Line 16: The statement the auroral particle precipitation *is due to* . . . does not make physical sense. This sentence needs to be reworded.

Page 2, Line 22: "over" a wide energy range, rather than "on" a wide energy range.

Page 6, Line 25: "Of course", rather than "Of cause". Although that is somewhat colloquial, and isn't really required in the sentence/

Page 8, Line 19: "A potential explanation", rather than "An explanation".

---

## Author Comment (AC1) · 28 Jul 2019

**Response to Reviewer letter**

Olesya Yakovchuk        Jan Maik Wissing

July 28, 2019

Manuscript Title: Magnetic local time asymmetries in precipitating and trapped electron and proton populations with and without substorm activity

https://www.ann-geophys-discuss.net/angeo-2019-49

https://doi.org/10.5194/angeo-2019-49-RC1

We would like to thank the reviewers for fruitful comments as they draw out attention to aspects that can be (better) answered by the used methods but somehow were off our radar.

As a couple of figures have been added (and one table has been replaced) we would like to note that the references in this reply belong to the old numbering.

**Reviewer 1 (Comments)**

The paper uses POES and METOP data to evaluate how the precipitation of protons and electrons varies with magnetic local time during both isolated substorms and in the absence of substorm activity. Data collected during the declining phase of cycle 23 has been used, and measurements from multiple energy channels evaluated. Overall this is an interesting piece of work and generally well written. The introduction and motivation in particular is very well thought out.

**1.  Comment**

My major concern for the paper is that one of the authors main results, the Kp (or lack of) dependence of proton and electron precipitation, is discussed in the abstract as well as the summary section, but any figure actually showing this has been omitted from the paper. While an indication is given in table 2, this is difficult to assimilate and results considering different Kp bins need to be presented as a figure for this result to be claimed.

**Reply:** *The reviewer is right, that table 2 is just an "indication" of the Kp dependency that we noticed. However, it already pointed out what we now elaborated in more detail. As all channels and various Kp level had to be analyzed, it was now necessary to write an automated auroral oval detection algorithm for APEX 110 km latitude or MLT locations of the auroral oval or its flux maximum and minimum. The routine determines the maximum flux for each MLT-bin within the typical auroral latitude range. This results in a preliminary auroral oval. Then the latitudinal differences between MLT-predecessor and successor are determined and in case of large outliers a point is assumed to be a spike in the data and replaced by the next biggest flux-bin in that MLT*

*zone. In case that more than 7 points have to be replaced for a auroral oval the according channel-Kp set is neglected. In sum this ends up in a well-working detection algorithm for the auroral oval and allows us to find its minimum and maximum flux, or their ratio. A sample output is given in the following figure.*

[Figure]

*The gray dots represent the position of the auroral*

oval. The green (9 MLT) and black (20 MLT) dots indicate maximum and minimum of the auroral oval, respectively. All locations have been cross-checked manually.

[Figure]

The next figure is based on these findings and presents the ratio between maximum and minimum auroral oval flux (or in other words the asymmetry of the oval) depending on Kp-level for every channel separately. Actually the channels have been grouped by their Kp dependency. The upper panel shows the 2 lowest electron channels and the lowest proton channel which all have a declining flux asymmetry with increasing Kp. The 6-6.7 Kp-bin is enhanced here, but we should keep in mind that this levels are occurring rarely and may suffer from bad statistics.
The middle panel shows all particle channels that have an increasing flux asymmetry with Kp, as they are: all remaining electron channels and the proton channels TED band 11 and mep0P1.
The lowest panel gives the flux asymmetry dependencies of the remaining proton channels that are less distinct. It seems that there is a domain change at about 3.3 Kp,

since the asymmetry of TED proton band 14 and mep0P2 has a negative correlation below 3.3 and a positive one above. For the channels TED proton band 8 and mep0P3 the relationship is opposite.

All these findings are based on the whole period including all (no-) substorm conditions.
Another aspect that was mentioned in table 2 is how the asymmetry depends on substorms. The next figure presents this relation in more detail. Since an 8 year period does not contain enough values for substorms in rare Kp-levels we neglected the Kp-level here and compared isolated substorm to no-substorm periods.

[Figure]

Except for TED electron band 4 (where there is no significant difference between substorm and no-substorm periods) all other channels have an increased auroral oval asymmetry during isolated substorms. The numbers above and below the marked flux ratio indicate the MLT location of the minimum (below) and maximum (above). We can identify that the maximum flux during a substorm shifts to the midnight sector (if not already there in no-substorm periods) e.g. for mep0e1-e2, TED proton band 4 and 14.
For TED electron band 4 and 8 (as well as TED proton band 8 and 11) the substorm enhancement is also seen in the night sector, but it does not overshoot the dayside flux (see Figures 2 and 3), while the substorm enhancement in the night-sector of mep0e2-e3 is in the same order as the 9–12 MLT flux.
The asymmetry in both, the electron and the proton spectrum shows a local minimum in middle TED channels (TED electron band 8 and proton band 11) as well as a local maximum at higher energies (TED electron band 14 or mep0e1-e2 for electron and mep0P1 or mep0P2 for protons). At even higher energies the asymmetry de-

*clines again.*

*This information has been added to the paper. Sections have been restructured accordingly. Given that the new figures are more detailed than the previous table, the table will be omitted.*

**2. Comment**

Line 19, section 1, discusses how the SML index was used to define the substorm onsets. While the reviewer agrees that SML is a good choice to define substorm onsets, perhaps the authors could elaborate on why SML was used instead of the AE index as in the Reeves et al., 2003 study.

**Reply:** *The link between AE and SML now is explained in more detail. The paragraph now reads: The occurrence of substorms depends on the orientation of the interplanetary magnetic field (Reeves et al., 2003). As shown in Reeves et al. (2003) these external solar wind parameters subsequently impact the magnetic field on the ground and are represented in the Auroral Electrojet (AE) index. Auroral Electrojet indices AE=AU-AL are a good proxy of the global auroral power, where AU and AL are the upper and lower components of AE, which means the largest and smallest values of the H component among 12 magnetic stations (Davis und Sugiura, 1966). AU represents the strength of the eastward electrojet, while AL represents the westward electrojet. Consequently AL seems to be the index which best corresponds to westward intensification of the auroral current aka substorm activity. Prior to substorm onset, AL index is typically small in magnitude, with the contributing station near dawn, whereas during substorm onset, the station contributing to the lower envelope is usually in the dusk sector under the auroral expansion. However, due to the limited spatial coverage of the 12 magnetometer stations the auroral expansion can be missed, which means that this index does not always reflect the onset (Gjerloev et al., 2004). The use of SuperMAG SML, an index derived likewise to the AE but based on all available magnetometer stations (typically more than 100) at these latitudes, considerably improves the detection of substorm onsets (Newell und Gjerloev, 2011). Thus we use the SML index in this study to define substorm onsets.*

**3. Comment**

In section 3, line 11-12 reads "the SAA allows energetic particles in the radiation belt to reach altitudes low enough to be reached by the satellites orbit". Considering that the authors are solely using the T0 flux measured by POES and METOP, Figure 1 in Rodger et al. (2010) would suggest that, even over the SAA, the T0 measured flux is still precipitating.

**Reply:** *Comparing Figure 1 in Rodger et al. (2010) to the upper panel of our Fig. 1 (in geographic coordinates) we can see that the SAA region (between 280 and 360 degrees East and -45 to 0 degrees North) covers a mix of all populations. The central part of the SAA is even located in the yellow area labled with "trapped+drift loss cone+bounce loss cone". In so far we disagree with the reviewer and conclude that the T0 flux measured in the SAA is not precipitating in total since it also contains a fraction of the trapped particles. As the following sentence reads "Thus the high flux values are not necessarily connected to high particle precipitation." we already tried to mention that. However, to point that out, we now added the following sentence: "According to Rodger et al. (2010: Fig. 1) the particle population in the SAA consists of particles precipitating in bounce and drift loss cone as well as trapped particles."*

**4. Comment**

In Figures 2, 3, and 4, as well as in the text, could the different energy channels be referred to by the energy range covered rather than the channel name? This would make the results easier to interpret without constantly flicking back to table 1.

**Reply:** *We agree with the reviewer that it might be helpful to give the energy ranges in every figure. However there are some caveats about the energy ranges that need to be mentioned: a) Some channels suffer from degradation. This mostly holds for the MEPED proton channels and is a result of structural defects caused by the impinging particles. On the long run it causes an energy shift (to higher particles energies) since less electron-hole pairs are produced per deposited particle energy. Consequently the mentioned energy ranges are nominal ranges. Further details on degradation of the MEPED channels can be found in e.g. Asikainen et al. (2012). b) For at least one channel the energy range seems to be doubtful as NOAA describes the same detectors in two technical documents with divergent ranges for mep0P2: 80–240 keV in (Evans und Greer, 2006) and the same channel in Green (2013): 80–250 keV. The electron channels also have different ranges as Green (2013) lacks an upper threshold energy. But since we subtract the electron channels in order to get differential channels this does not matter in our case anyway.*

*We added the nominal energy range to all figures and added a note about the possible degradation.*

**5. Comment**

When discussing figure 2 in section 4, point c and d mention that the noon sector flux decreases during a substorm. Could the authors speculate on why this is?

**Reply:** *We added the following information: "The noon sector flux decreases most probably because dayside particle precipitation occurs often during northward oriented IMF which is not usual for substorms."*

in the day-sector and thus at significantly lower latitudes. This information has been added to the paper.

**6. Comment**

When discussing Figure 3, the first point claims that protons also show an equatorial movement of the main precipitation zone with increasing particle energy. More direction to this in the figure is required here for the reader as I do not see this.

**Reply:** *The following figure presents the dependency of the latitude of the main precipitation zone to the particle energy.*

[Figure]

*The figure has been derived by the auroral oval determination method discussed before and displays the latitude of maximum auroral oval flux. Except for some outliers, most of them belonging to TED proton band 4 during high Kp levels ($> 6$), the graphs show a clear equatorial dislocation with increasing energy. The 110 km APEX latitudinal range at a specific Kp-level is about $10°$ for electrons and $12-16°$ for protons. This dislocation however appears to be stepwise: TED electron band 4 and 8 are on the same latitude as well as TED electron band 14, mep0e1-e2 and mep0e2-e3 share the same latitude. For protons TED proton band 4, 8 and 11 are on one latitude and the higher particle energies mep0P1 and mep0P2 are colocated. This implies that these particles origin from the same souce region.*
*Every color graph represents the spectral location of the main precipitation zone for a certain Kp-range. Thus we can infer that increased geomagnetic disturbance (high Kp-values) causes a dislocation of up to about $8°$ towards the equator.*
*Concerning the outliers in TED proton band 4, for low Kp-values there is a clear flux maximum at noon, which is located at rather high latitudes (compare Fig. 2). At high Kp, the MLT asymmetry declines and then flips. Consequently the maximum flux for high Kp-levels is not*

**7. Comment**

The second point made when discussing Figure 3 was that there is a second auroral oval. This is then stated to be an artefact of the MLT binning. Could the authors add further explanation to justify this here?

**Reply:** *We added a figure tha shows the APEX latitude and longitude. This figure nicely shows that the double auroral structure is on the same latitude as the drift loss cone. Also further explanation has been added. The following information now is included: "Figure 1 shows the TED proton band 11 in geographic coordinates (top row) and modified APEX 110 km coordinates (bottom row). The left column shows latitude against longitude while the right column shows latitude against MLT. No selection according to Kp-level or substorm intensity has been made, while all available data from METOP 2 and POES 15, 16, 17 and 18 for the years 2001–2008 has been included. This allows a spatial resolution of 3.75 degrees longitude (or 15 min MLT). Please note that latter figures show a reduced longitudinal resolution of 15 degrees (or 1 h MLT) only to avoid statistical noise in e.g isolated substorms periods."*
*and*
*"Figure 1 (top, right) shows the same data on a geographic latitude vs. MLT grid. As the auroral oval is not visible as an oval any more but mixes up local time differences and the latitudinal variations that can already be seen in Fig. 1 (top, left)."*
*and*
*"Switching to magnetic modified APEX 110 km coordinates (see Fig 1, bottom, left) straightens the auroral oval and mostly removes the longitudinal dependence except for the SAA and the drift loss cone in the South of the SAA. Consequently we can replace the APEX longitude by MLT (see Fig. 1, bottom, right). Features that depend on magnetic local time now become visible and the auroral oval itself does not show a hemispheric dependence. The SAA and the drift loss cone, however, are now smeared out and still produce a hemispheric asymmetry. The drift loss cone that is located at a distinct modified APEX 110 km longitudinal range even appears as an double auroral structure at the same latitude but covering all longitudes. Which of course is an artifact of this kind of MLT binning."*
*In addition several references to the figure have been updated.*

**8. Comment**

In the text, table 3 is not discussed or explained. Perhaps it is not needed? Otherwise the relevance needs to be discussed.

**Reply:** *The table is now referenced in the text.*

**Typo**

Additionally, there are a number of typographical errors in the manuscript:

Page 4, line 3: "furtunately",
**Reply:** *Corrected.*

**Typo**

Page 4, line 15: "trapped particles in low altitudes" should be "at low altitudes".

**Reply:** *Assuming that latitudes are meant we corrected the wrong preposition throughout the manuscript.*

**Typo**

Page 4, Line 25: "mainly located in about 110 km altitude" should be "at about 110 km...".

**Reply:** *Corrected.*

**Typo**

Page 4, Line 34: you have not defined the QD acronym - please write out quasi-dipole.

**Reply:** *The acronym has been added were the quasi-dipole system is introduced.*

**Typo**

Spelling of "avoid" throughout the manuscript is wrong.
**Reply:** *Corrected.*

**Typo**

Page 5, line 20: "independend".
**Reply:** *Corrected to "independent".*

**Typo**

Page 6, line 29: "Substorm depended precipitation..."
**Reply:** *Corrected.*

**References**

Asikainen, T. ; Mursula, K. ; Maliniemi, V.: Correction of detector noise and recalibration of NOAA/MEPED energetic proton fluxes. In: *J. Geophys. Res.* 117 (2012), S. A09204. `http://dx.doi.org/doi:10.1029/2012JA017593`. – DOI doi:10.1029/2012JA017593.

Davis, T. N. ; Sugiura, M.: Auroral electrojet activity index AE and its universal time variations. In: *J. Geophys. Res.* 71 (1966), Februar, S. 785–801. `http://dx.doi.org/10.1029/JZ071i003p00785`. – DOI 10.1029/JZ071i003p00785.

Evans, D. S. ; Greer, M. S.: *Polar Orbiting Environmental Satellite Space Environment Monitor - 2, Instrument Descriptions and Archive Data Documentation.* NOAA Space Environ. Lab, Boulder, Colorado, USA: National Oceanic and Atmospheric Administration, 2006. – 51 S. – Version 2.0.

Gjerloev, J. ; Hoffman, R. ; Friel, M. ; Frank, L. ; Sigwarth, J.: Substorm behavior of the auroral electrojet indices. In: *Annales Geophysicae* 22 (2004), Juni, S. 2135–2149. `http://dx.doi.org/10.5194/angeo-22-2135-2004`. – DOI 10.5194/angeo–22–2135–2004.

Green, Janet: NOAA NESDIS-NGDC MEPED Telescope Data Processing ALGORITHM THEORETICAL BASIS / NOAA National Geophysical Data Center. 2013. – Forschungsbericht. – Version 1.0.

Newell, P. T. ; Gjerloev, J. W.: Evaluation of SuperMAG auroral electrojet indices as indicators of substorms and auroral power. In: *Journal of Geophysical Research (Space Physics)* 116 (2011), Dezember, S. A12211. `http://dx.doi.org/10.1029/2011JA016779`. – DOI 10.1029/2011JA016779.

Reeves, G. D. ; Henderson, M. G. ; Skoug, R. M. ; Thomsen, M. F. ; Borovsky, J. E. ; Funsten, H. O. ; C:son Brandt, P. ; Mitchell, D. J. ; Jahn, J.-M. ; Pollock, C. J. ; McComas, D. J. ; Mende, S. B.: IMAGE, POLAR, and Geosynchronous Observations of Substorm and Ring Current Ion Injection. In: Surjalal

Sharma, A. (Hrsg.) ; Kamide, Y. (Hrsg.) ; Lakhina, G. S. (Hrsg.): *Disturbances in Geospace: The Storm-substorm Relationship* Bd. 142, 2003 (Washington DC American Geophysical Union Geophysical Monograph Series), S. 91.

Rodger, Craig J. ; Clilverd, Mark A. ; Green, Janet C. ; Lam, Mai M.: Use of POES SEM-2 observations to examine radiation belt dynamics and energetic electron precipitation into the atmosphere. In: *Journal of Geophysical Research: Space Physics* 115 (2010), Nr. A4. `http://dx.doi.org/10.1029/2008JA014023`. – DOI 10.1029/2008JA014023.

---

## Author Comment (AC2) · 13 Aug 2019

**Response to Reviewer letter**

Olesya Yakovchuk          Jan Maik Wissing

August 13, 2019

Manuscript Title: Magnetic local time asymmetries in precipitating and trapped electron and proton populations with and without substorm activity

https://www.ann-geophys-discuss.net/angeo-2019-49
https://doi.org/10.5194/angeo-2019-49-RC1

**Annotation**

We would like to thank the reviewers for fruitful comments as they draw out attention to aspects that can be (better) answered by the used methods but somehow were off our radar.

As a couple of figures have been added (and one table has been replaced) we would like to note that the references in this reply belong to the old numbering.

Additionally we ereased the section about low latitudes since we noticed that the different longitudes contribute very unevenly to the MLT bins here. In case that reviewers' comments to that section have been answered before, we included the answers. Further explanation is given in a spectial section that reads:

**1. Why is the SAA not evenly smeared out over all longitudes?**

If we would take a look at the footpoints of a solar-synchronous satellite in local time we would see that it always crosses a particular latitude e.g. the equator at one particular local time in ascending mode (and another, at the equator 12 hours later, in descending mode). At high latitudes it crosses 12 local time zones on a few latitudes, but still, the next orbit will exactly match the first (except if the orbit moves, which also happens to the POES/METOP satellites, but on longer time scales). Looking at the footpoints of the same satellite in MLT changes quite a bit. Given that the MLT zones are based on magnetic latitude and the magnetic poles being shifted, it means that the MLT-footpoints especially in high latitudes differ significantly from one orbit to the next. Due to the POES inclination of 98.5 degrees the satellite may at maximum reach the northern magnetic pole. The southern magnetic pole however, may not only be reached but even passed.

Thus there are two options how the MLT during an orbit may develop in the southern hemisphere: If the satellite's longitude is far from the magnetic pole the orbit will not pass the magnetic pole and the MLT will gradually increase by 12 hours till it reaches the equator in ascending mode again. Let us call this "ascending MLT". In the other case ("descending MLT"), the southern magnetic pole will be passed and the magnetic local time zones will be flown through in the opposite direction, decreasing MLT by 12 hours till reaching the equator in ascending mode again. Since the southern magnetic pole is somewhat south of Australia a significant fraction of the orbits passing it will cross the SAA in descending mode (but not in ascending mode). The opposite is true for the ascending MLT path, which includes a significant fraction of orbits that pass the SAA in ascending mode.

In case multiple satellite are used this does not affect high latitudes, but in low latitudes the situation is different. Since the satellites are crossing the equator at two specific local times (for ascending and descending mode, being just slightly broader in MLT), MLT coverage at the equator is limited to these points. They however may be reached in ascending mode (or left in descending mode) by ascending or descending MLT paths. In Fig. 1, top right (or bottom right) the equator is crossed at six different smeared out MLTs. While the ones on the left (13, 17 -two satellites- and 21 MLT) represent the descending mode, the ones on the right (1, 5 and 9 MLT) are in ascending mode.

The ascending MLT path now connects e.g. the low flux right edge of the 21 MLT equatorial crossing with the high flux (SAA) left edge of the 9 MLT equatorial crossing. The descending MLT path on the other side connects e.g. the high flux (SAA) left edge of the 21 MLT equatorial crossing with the low flux right edge of the 9 MLT equatorial crossing.

In sum the ascending and descending MLT paths cause the left edge of an equatorial crossing to be affected by the SAA, while the right edge is not. Any MLT analysis of low latitudes based on POES/MEPED will suffer from the fact that the longitudes contribute very unevenly to the MLT zones. This hampers a flux analysis in low latitudes. In high latitudes however this effects gets counterbalanced by broader MLT coverage and multiple satellites.

**Reviewer 1 (Comments)**

The paper uses POES and METOP data to evaluate how the precipitation of protons and electrons varies with magnetic local time during both isolated substorms and in the absence of substorm activity. Data collected during the declining phase of cycle 23 has been used, and measurements from multiple energy channels evaluated. Overall this is an interesting piece of work and generally well written. The introduction and motivation in particular is very well thought out.

**1. Comment**

[Figure]

My major concern for the paper is that one of the authors main results, the Kp (or lack of) dependence of proton and electron precipitation, is discussed in the abstract as well as the summary section, but any figure actually showing this has been omitted from the paper. While an indication is given in table 2, this is difficult to assimilate and results considering different Kp bins need to be presented as a figure for this result to be claimed.

**Reply:** *The reviewer is right, that table 2 is just an "indication" of the Kp dependency that we noticed. However, it already pointed out what we now elaborated in more detail. As all channels and various Kp level had to be analyzed, it was now necessary to write an automated auroral oval detection algorithm for APEX 110 km latitude or MLT locations of the auroral oval or its flux maximum and minimum. The routine determines the maximum flux for each MLT-bin within the typical auroral latitude range. This results in a preliminary auroral oval. Then the latitudinal differences between MLT-predecessor and successor are determined and in case of large outliers a point is assumed to be a spike in the data and replaced by the next biggest flux-bin in that MLT zone. In case that more than 7 points have to be replaced for a auroral oval the according channel-Kp set is neglected. In sum this ends up in a well-working detection algorithm for the auroral oval and allows us to find its minimum and maximum flux, or their ratio. A sample output is given in the following figure.*

*The gray dots represent the position of the auroral oval. The green (9 MLT) and black (20 MLT) dots indicate maximum and minimum of the auroral oval, respectively. All locations have been cross-checked manually.*

[Figure]

[Figure]

The next figure is based on these findings and presents the ratio between maximum and minimum auroral oval flux (or in other words the asymmetry of the oval) depending on Kp-level for every channel separately. Actually the channels have been grouped by their Kp dependency. The upper panel shows the 2 lowest electron channels and the lowest proton channel which all have a declining flux asymmetry with increasing Kp. The 6-6.7 Kp-bin is enhanced here, but we should keep in mind that this levels are occurring rarely and may suffer from bad statistics.

The middle panel shows all particle channels that have an increasing flux asymmetry with Kp, as they are: all remaining electron channels and the proton channels TED band 11 and mep0P1.

The lowest panel gives the flux asymmetry dependencies of the remaining proton channels that are less distinct. It seems that there is a domain change at about 3.3 Kp, since the asymmetry of TED proton band 14 and mep0P2 has a negative correlation below 3.3 and a positive one above. For the channels TED proton band 8 and mep0P3

the relationship is opposite.

All these findings are based on the whole period including all (no-) substorm conditions.

Another aspect that was mentioned in table 2 is how the asymmetry depends on substorms. The next figure presents this relation in more detail. Since an 8 year period does not contain enough values for substorms in rare Kp-levels we neglected the Kp-level here and compared isolated substorm to no-substorm periods.

Except for TED electron band 4 (where there is no significant difference between substorm and no-substorm periods) all other channels have an increased auroral oval asymmetry during isolated substorms. The numbers above and below the marked flux ratio indicate the MLT location of the minimum (below) and maximum (above). We can identify that the maximum flux during a substorm shifts to the midnight sector (if not already there in no-substorm periods) e.g. for mep0e1-e2, TED proton band 4 and 14.

For TED electron band 4 and 8 (as well as TED proton band 8 and 11) the substorm enhancement is also seen in the night sector, but it does not overshoot the dayside flux (see Figures 2 and 3), while the substorm enhancement in the night-sector of mep0e2-e3 is in the same order as the 9–12 MLT flux.

The asymmetry in both, the electron and the proton spectrum shows a local minimum in middle TED channels (TED electron band 8 and proton band 11) as well as a local maximum at higher energies (TED electron band 14 or mep0e1-e2 for electron and mep0P1 or mep0P2 for protons). At even higher energies the asymmetry declines again.

This information has been added to the paper. Sections have been restructured accordingly. Given that the new

*figures are more detailed than the previous table, the table will be omitted.*

**2. Comment**

Line 19, section 1, discusses how the SML index was used to define the substorm onsets. While the reviewer agrees that SML is a good choice to define substorm onsets, perhaps the authors could elaborate on why SML was used instead of the AE index as in the Reeves et al., 2003 study.

    **Reply:** *The link between AE and SML now is explained in more detail. The paragraph now reads: The occurrence of substorms depends on the orientation of the interplanetary magnetic field (Reeves et al., 2003). As shown in Reeves et al. (2003) these external solar wind parameters subsequently impact the magnetic field on the ground and are represented in the Auroral Electrojet (AE) index. Auroral Electrojet indices AE=AU-AL are a good proxy of the global auroral power, where AU and AL are the upper and lower components of AE, which means the largest and smallest values of the H component among 12 magnetic stations (Davis und Sugiura, 1966). AU represents the strength of the eastward electrojet, while AL represents the westward electrojet. Consequently AL seems to be the index which best corresponds to westward intensification of the auroral current aka substorm activity. Prior to substorm onset, AL index is typically small in magnitude, with the contributing station near dawn, whereas during substorm onset, the station contributing to the lower envelope is usually in the dusk sector under the auroral expansion. However, due to the limited spatial coverage of the 12 magnetometer stations the auroral expansion can be missed, which means that this index does not always reflect the onset (Gjerloev et al., 2004). The use of SuperMAG SML, an index derived likewise to the AE but based on all available magnetometer stations (typically more than 100) at these latitudes, considerably improves the detection of substorm onsets (Newell und Gjerloev, 2011). Thus we use the SML index in this study to define substorm onsets.*

**3. Comment**

In section 3, line 11-12 reads "the SAA allows energetic particles in the radiation belt to reach altitudes low enough to be reached by the satellites orbit". Considering that the authors are solely using the T0 flux measured by POES and METOP, Figure 1 in Rodger et al. (2010) would suggest that, even over the SAA, the T0 measured flux is still precipitating.

    **Reply:** *Comparing Figure 1 in Rodger et al. (2010) to the upper panel of our Fig. 1 (in geographic coordinates) we can see that the SAA region (between 280 and 360 degrees East and -45 to 0 degrees North) covers a mix of all populations. The central part of the SAA is even located in the yellow area labled with "trapped+drift loss cone+bounce loss cone". In so far we disagree with the reviewer and conclude that the T0 flux measured in the SAA is not precipitating in total since it also contains a fraction of the trapped particles. As the following sentence reads "Thus the high flux values are not necessarily connected to high particle precipitation." we already tried to mention that. However, to point that out, we now added the following sentence: "According to Rodger et al. (2010: Fig. 1) the particle population in the SAA consists of particles precipitating in bounce and drift loss cone as well as trapped particles."*

**4. Comment**

In Figures 2, 3, and 4, as well as in the text, could the different energy channels be referred to by the energy range covered rather than the channel name? This would make the results easier to interpret without constantly flicking back to table 1.

    **Reply:** *We agree with the reviewer that it might be helpful to give the energy ranges in every figure. However there are some caveats about the energy ranges that need to be mentioned: a) Some channels suffer from degradation. This mostly holds for the MEPED proton channels and is a result of structural defects caused by the impinging particles. On the long run it causes an energy shift (to higher particles energies) since less electron-hole pairs are produced per deposited particle energy. Consequently the mentioned energy ranges are nominal ranges. Further details on degradation of the MEPED channels can be found in e.g. Asikainen et al. (2012). b) For at least one channel the energy range seems to be doubtful as NOAA describes the same detectors in two technical documents with divergent ranges for mep0P2: 80–240 keV in (Evans und Greer, 2006) and the same channel in Green (2013): 80–250 keV. The electron channels also have different ranges as Green (2013) lacks an upper threshold energy. But since we subtract the electron channels in order to get differential channels this does not matter in our case anyway.*
*We added the nominal energy range to all figures and added a note about the possible degradation.*

**5. Comment**

When discussing figure 2 in section 4, point c and d mention that the noon sector flux decreases during a substorm. Could the authors speculate on why this is?

    **Reply:** *We added the following information: "The noon sector flux decreases most probably because dayside*

*particle precipitation occurs often during northward orientated IMF which is not usual for substorms."*

**6.   Comment**

When discussing Figure 3, the first point claims that protons also show an equatorial movement of the main precipitation zone with increasing particle energy. More direction to this in the figure is required here for the reader as I do not see this.

**Reply:** *The following figure presents the dependency of the latitude of the main precipitation zone to the particle energy.*

[Figure]

*The figure has been derived by the auroral oval determination method discussed before and displays the latitude of maximum auroral oval flux. Except for some outliers, most of them belonging to TED proton band 4 during high Kp levels ($> 6$), the graphs show a clear equatorial dislocation with increasing energy. The 110 km APEX latitudinal range at a specific Kp-level is about $10°$ for electrons and $12\text{-}16°$ for protons. This dislocation however appears to be stepwise: TED electron band 4 and 8 are on the same latitude as well as TED electron band 14, mep0e1-e2 and mep0e2-e3 share the same latitude. For protons TED proton band 4, 8 and 11 are on one latitude and the higher particle energies mep0P1 and mep0P2 are colocated. This implies that these particles origin from the same souce region.*

*Every color graph represents the spectral location of the main precipitation zone for a certain Kp-range. Thus we can infer that increased geomagnetic disturbance (high Kp-values) causes a dislocation of up to about $8°$ towards the equator.*

*Concerning the outliers in TED proton band 4, for low Kp-values there is a clear flux maximum at noon, which is located at rather high latitudes (compare Fig. 2). At high Kp, the MLT asymmetry declines and then flips. Consequently the maximum flux for high Kp-levels is not in the day-sector and thus at significantly lower latitudes. This information has been added to the paper.*

**7.   Comment**

The second point made when discussing Figure 3 was that there is a second auroral oval. This is then stated to be an artefact of the MLT binning. Could the authors add further explanation to justify this here?

**Reply:** *We added a figure tha shows the APEX latitude and longitude. This figure nicely shows that the double auroral structure is on the same latitude as the drift loss cone. Also further explanation has been added. The following information now is included: "Figure 1 shows the TED proton band 11 in geographic coordinates (top row) and modified APEX 110 km coordinates (bottom row). The left column shows latitude against longitude while the right column shows latitude against MLT. No selection according to Kp-level or substorm intensity has been made, while all available data from METOP 2 and POES 15, 16, 17 and 18 for the years 2001–2008 has been included. This allows a spatial resolution of 3.75 degrees longitude (or 15 min MLT). Please note that latter figures show a reduced longitudinal resolution of 15 degrees (or 1 h MLT) only to avoid statistical noise in e.g isolated substorms periods."*
*and*
*"Figure 1 (top, right) shows the same data on a geographic latitude vs. MLT grid. As the auroral oval is not visible as an oval any more but mixes up local time differences and the latitudinal variations that can already be seen in Fig. 1 (top, left)."*
*and*
*"Switching to magnetic modified APEX 110 km coordinates (see Fig 1, bottom, left) straightens the auroral oval and mostly removes the longitudinal dependence except for the SAA and the drift loss cone in the South of the SAA. Consequently we can replace the APEX longitude by MLT (see Fig. 1, bottom, right). Features that depend on magnetic local time now become visible and the auroral oval itself does not show a hemispheric dependence. The SAA and the drift loss cone, however, are now smeared out and still produce a hemispheric asymmetry. The drift loss cone that is located at a distinct modified APEX 110 km longitudinal range even appears as an double auroral structure at the same latitude but covering all longitudes. Which of course is an artifact of this kind of MLT binning."*
*In addition several references to the figure have been updated.*

**8.   Comment**

In the text, table 3 is not discussed or explained. Perhaps it is not needed? Otherwise the relevance needs to be discussed.

**Reply:** *The table is now referenced in the text.*

**Typo**

Additionally, there are a number of typographical errors in the manuscript:

Page 4, line 3: "furtunately",
**Reply:** *Corrected.*

**Typo**

Page 4, line 15: "trapped particles in low altitudes" should be "at low altitudes".
**Reply:** *Assuming that latitudes are meant we corrected the wrong preposition throughout the manuscript.*

**Typo**

Page 4, Line 25: "mainly located in about 110 km altitude" should be "at about 110 km...".
**Reply:** *Corrected.*

**Typo**

Page 4, Line 34: you have not defined the QD acronym - please write out quasi-dipole.
**Reply:** *The acronym has been added were the quasi-dipole system is introduced.*

**Typo**

Spelling of "avoid" throughout the manuscript is wrong.
**Reply:** *Corrected.*

**Typo**

Page 5, line 20: "independend".
**Reply:** *Corrected to "independent".*

**Typo**

Page 6, line 29: "Substorm depended precipitation..."
**Reply:** *Corrected.*

**Reviewer 2 (Comments)**

(We reordered the reviewer's comments to allow more structured answering.)

The paper describes electron and proton differential number flux $1/(m^2$-s-sr-MeV- according to the figure labels) measured in various energy channels by detectors on POES and METOP from 2001-2008 in the radial direction away from Earth for electrons and 9 degrees from that in protons. It presents the results of these by field line mapped latitude at 110 km altitude and by magnetic local time (MLT) for non-sub-storm times and isolated sub-storms. It also makes mention of comparisons to geomagnetic activity (Kp), but does not present any data in this regard. The data in the paper could have potential contributions to the field with additional work and resubmission.

At a very minimum these two major issues need to be addressed: the data and results need to be presented within the proper context (a useful review of the TED energy range precipitation is in by Frey, 2007), and a more precise description of what is presented/covered in the paper is required.

Until the paper is reworked into the proper context, it is difficult to determine whether the conclusions are significant (currently they are not), how important the contribution is, or if the length of the paper is adequate. The language is mostly fluent, but does need to be made more precise in ways, as mentioned. The figures are of adequate size, although the text in them is too small in some cases. The presentation is clear and organized, although missing the context for appropriate organization.

**1. Comment**

The data itself is a contribution to the field providing the same type of measurements and analysis over a wide energy range for both electrons and protons. However, the difference in scales for the different channels and the substantial difference in energy widths of the channels makes inter-comparison between the channels difficult other than relative locations of peaks and troughs and differentials, which are the focus of the paper results. Multiplying by the geometric mean of the channel energy ranges and dividing by the channel energy width would give a reasonable proxy for energy flux, which could be put on the same scale and would allow direct comparison of levels between the energy channels, which would add to the contribution from the presentation of the data.

**Reply:** *We agree with the reviewer that dividing the particle flux by the channels' energy range (resulting in a so called "differential flux" given in $1/(m^2\,s\,sr\,MeV)$, which we are using in the paper) and multiplying it with the mean kinetic energy of a particle detected in those channels would result in values that are closer together than the differential flux itself. However, as acceleration mechanisms are different, we do not see that "closer together" is equivalent to "more comparable". Our intention for the inter-comparison is two-fold: First we are interested in asymmetries in the main precipitation zone, the auroral oval. Using the minimum flux in the auroral*

*oval as reference allows us to easily identify and quantify these asymmetries by color and in so far it sounds reasonable for us. Second, the differential flux is needed when transposing the particle precipitation into atmospheric ionization. This is not part of this paper, but since a later paper about atmospheric ionization will refer to these asymmetries it might be a good idea to have them in the correct format already.*

*We now added "All particle count rates have been converted into differential flux by dividing the energy range and a geometric factor has been applied as suggested in Evans und Greer (2006)." to avoid any misinterpretation. Also the paragraph about satellite data has been restructured.*

**2. Comment**

The data would be a contribution (and could be improved as indicated), but a good amount of additional is required to other facets of the paper. In particular, there are two major components that need to be significantly improved. One is the need to put the data in appropriate context of previous work, empirical and physical theories. The results also would need to be appropriately put into context. The second is a loose use of language in important ways that causes misrepresentation of the data and focus of the paper.

In particular, the has been a substantial amount of work done of the latitude and MLT distributions of electron (and proton) precipitation, particularly at auroral latitudes and at the lower (TED equivalent) energies, such as Cattell et al, 2006, Newell et al, 2009, and more recently, Dombeck et al, 2018 and the references within these. There is also a substantial number of papers on the MLT and latitude dependence and relationship between Alfvén waves and electron precipitation, by the likes of Keiling, Chaston and Hatch. Although these later (Alfvén wave papers) only relate to one small energy band and one part of the precipitation in the channel, so comparison to the presented results would be indirect, so perhaps could not be included in detail. However, the Cattell, Newell and Dombeck papers all show latitude versus MLT plots of precipitation that can and should be compared to the data and results presented in this paper.

**Reply:** *A couple of comparisons especially with Newell et al. (2009) have been included now. However, the different treatment of energy channels in our paper and in Newell et al. (2009), Dombeck et al. (2018) and Cattell et al. (2006) papers makes a comparison difficult. We are looking at different channels (equivalent to energies) while these authors are looking at number flux*

*and energy flux. Of course number flux and energy flux can be interpreted as low energies and high energies, but a direct one-to-one comparison is not possible. Additionaly, as the reviewer anncounces, it holds only for a very small fraction of our energy spectrum.*

**3. Comment**

To get precipitation (other than drift lose cone affects in the SAA, which the authors are pointedly not trying to investigate). acceleration and/or wave scattering into the loss cone is necessary. These mechanisms have been discussed and investigated in the literature for decades, and they are dramatically different for the different species and energy channels discussed. All of this has direct relevance to the presented data and the proper and useful interpretation of the results from those data, and the data and results should be discussed from firmly within that context.

**Reply:** *The following paragraph has been added: "However, it is known that anisotropic distributions occur. While an unaccelerated source population is assumed to be isotropic (as is a wave-scattered fraction of that population in the loss cone) most acceleration processes are connected with an anisotropic pitch angle distribution. Dombeck et al. (2018) lists the most important ones as quasi-static-potential-structures, namely an electric potential field, that may cause isotropic or anisotropic distributions and Alfvén waves, that accelerate only particle energies that are in resonance with magnetic field wave and causes highly anisotropic distributions. Alfvén waves are responsible for electron precipitation during substorms (Newell et al., 2010). According to Newell et al. (2009) electrons are often accelerated while ions are not."*

**4. Comment**

For the second major issue, the loose use of language, the most egregious issue is the use of the loose use of the word precipitation. In particular, the title and abstract indicates that the paper is about precipitation, but even in the best case this is only half of the presented results, the highest latitude results. As the authors state, the lower latitudes are primarily measuring trapped populations. Therefore the title does not properly describe the paper. Even at the higher latitudes, what is being measured is the downgoing population (centered at various pitch angles depending on latitude, population, and relative potion of the spacecraft to Earth's magnetic pole). While this is still mostly "downward", downward does not necessarily mean precipitating, as the authors point out in relation to their discussion of the SAA. The authors need to be clear what it is that the data presented

are. The title and discussion should be adjusted accordingly.

**Reply:** *The reviewer is right that we focussed on the precipitation particles at the cost of trapped ones, especially now with erasing the low latitudes section. Our intention was to have the word "precipitation" in the title as it is the main buzz-word for someone who is also doing atmospheric ionization (as we do). Without the low latitude section the title should be more accurate.*
*We also agree that the particle flux measured at high latitudes is (without any assumptions) "downgoing" or "downward", while "precipitating" indicates that this flux reaches the top of the atmosphere. In fact considering "downgoing" and "precipitating" as equal -what we did, and which is probably the reviewer's critical point- is possible only when using the assumption of an isotropic pitch angle distribution. If assuming such an isotropical pitch angle distribution the flux decline due to mirroring particles and the flux increase due to a focussing flux tube balance each other (see e.g. Bornebusch et al., 2010) and thus the measured downgoing flux is equal to the precipitating flux. The paragraph now reads: "All figures in this paper are showing differential particle flux in $1/(MeV\, m^2\, s\, sr)$ as measured, thus we made no assumption about a pitch angle distribution here. However, it should be noted that even if the detector is looking upward (and measuring downgoing particles in high latitudes) it does not necessarily mean that all these particles are precipitating (reaching the atmosphere). Given that some particles are mirroring above the atmosphere a fraction of the downgoing flux is lost, thus the magnetic flux tube is narrowing the particle flux increases again and only in case that the pitch angle distribution is isotrotropic the mirrored fraction is balanced by flux tube narrowing (see e.g. Bornebusch et al., 2010). And only in case of an isotropic pitch angle distribution it does not matter for upscaling which angles of the downgoing pitch angle distribution we are measuring: an isotropic pitch angle distribution may easily be integrated over $2\pi$ to estimate the total precipitating flux over all angles.*
*However, it is known that anisotropic distributions occur. And in that case an estimation of the total precipitating flux is not straight forward as first a pitch angle distribution has to be assumed and second it has to be determined which pitch angles the detector is currently measuring. Since the only other detector orientation on POES is measuring trapped particles (at high latitudes) and since trapped particles do not get lost, there is no reason to assume a smooth transition between these two particle populations. Thus we do not have a "reference" anisotropic pitch angle distribution that might be applied. Applying an isotropic pitch angle (which is often done in literature) will put the downgoing flux on a level with*

*precipitating flux. In case that the paper states "particle precipitation" this isotropic pitch angle distribution has been implicitly assumed. Yet, this has been made without loss of generality since the shown differential flux in that case is equal to the downgoing flux. Thus no transformation is needed."*
*Additionaly we checked where the "particle precipitation" can be replaced by the more neural word "particle flux".*

**5. Comment**

Additional loose uses of language include discussions of "source" particles, and "qualitative" and "quantitative" results.
In particular, there are really only two sources of particles that are being detected: the solar wind/sun, and the ionosphere/Earth's atmosphere. One could argue that populations that are trapped in the ring current, plasmasphere, radiation belts, plasmasheet, etc, are different pools of particles that are "sources" for the particles measured in study. However, that argument has not been made in the paper. The authors also appear to using the word "source" to allude to both populations pool that the measured particles came from/belong to, and for the mechanisms that cause them to precipitate/be observed in the data. This really needs to be clearly described in the paper.

**Reply:** *We tried to describe it more clearly now.*

**6. Comment**

The authors also make a pointed distinction between "quantitative" and "qualitative" results in the paper. In nearly all cases, however, the results described in the paper are qualitative descriptions of the quantitative presented data. The only except to this is the minimum to maximum differences by MLT, which is purely quantitative. Regardless, the use of "qualitative" and "quantitative" as descriptors for the results is unnecessary and as used highly confusing (and inaccurate). As such they should likely just be removed.

**Reply:** *This section has been completely restructured with breaking up into subsections.*

**7. Comment**

Several other things that should be addressed in the next version of the paper, include the following: A figure showing coverage by MLT and latitude would be very helpful.

**Reply:** *This has been added in the new version of Figure 1 as well as in the according description, see also Comment 7. from Reviewer 1.*

**8.  Comment**

A thorough discussion of the minimum count levels of the instruments, and how the noise associated with 1 count levels, interpretation of zero counts, and differences between instruments and channels with regard to this are addressed should be included.

**Reply:** *The following information has been included: "All shown values are spatial and temporal averaged fluxes. In case that a detector measures zero counts every time it crosses a specific position and at a certain condition this also enters the figures with zero flux (see e.g. Fig. 2, TED electron band 11, isolated substorm, -55 degrees modified APEX latitude at noon). Since the detector counts are transferred into flux the MEPED channels do not recognize flux less than 1 count per integration interval (equivalent to $1\,000\,000\,particles/(m^2 ssr)$, divided by the channels energy range). For the TED detector the transformation is similar but instead of a fixed number a calibration factor has to be applied for every channel and satellite. The calibrations are given in e.g. Evans und Greer (2004)."*

**9.  Comment**

The field of view of the detectors should be discussed as well as the effects this has in interpretation of the results, from both the perspective of mirroring particles as well as potential anisotropic distributions.

**Reply:** *We added this paragraph about field of view, particle populations and potential anisotropic distributions.*

*"The MEPED detectors have a field of view of $\pm 15$ degrees, while the TED detector has the following specifications according to Evans und Greer (2006): The field of view of the electron and proton 1000–20 000 eV detector systems are $1.5°$ by $9°$, half angles. The field of view of the 50–1000 eV electron detector system is $6.7°$ by $3.3°$, half angles. The field of view of the 50-1000 eV proton detector system is $6.6°$ by $8.7°$, half angles. Opening angles of the detector in combination with the position of the satellite determines which particle populations the detector is measuring. According to Rodger et al. (2010: Fig. 1) the MEPED 0-degree detector in latitudes discussed in Section 4("High Latitudes") measures particles in the bounce loss cone only. Given that the point of view of the TED detector is almost identical with the MEPED detector and the field of view is significantly smaller, Figure 1 in Rodger et al. (2010) can also be applied, keeping in mind that regions of overlapping particle populations will decline. Thus we can borrow the particle populations seen in the TED channels from the MEPED results. In sum: at high latitudes both detectors count precipitating particle flux while they detect mostly*

*trapped particles at low latitudes.*

*All figures in this paper are showing differential particle flux in $1/(MeV\,m^2\,s\,sr)$ thus we made no assumption about a pitch angle distribution. However, if the pitch angle distribution is isotropic the shown particle flux may easily be integrated over $2\pi$ to estimate the total precipitating flux over all angles. But it is known that anisotropic distributions occur. In that case an estimation of the total precipitating flux is not straight forward as first a pitch angle distribution has to be assumed and second it has to be determined which pitch angles the detector is currently measuring."*

**10.  Comment**

The word "moves" is used to described the differences between energy channel results. This is not the appropriate word, they are different populations, the features are not "moving" in any sense.

**Reply:** *Well, if a graph shows a strong increase it is also possible to to describe it with "jumps up" even though it does probably not describe a kangaroo. We tried to avoid "move" and replaced it by "is shifted to", "is displaced to" or "is located at".*

**11.  Comment**

The "precipitation zone" is not at lower latitude with higher energy in general as indicated in the paper. The peak latitude appears identical for TED 8, 11, and 14, for example. The peak "precipitation zone" latitude is related to where particles of these energies reside (radiation belts, etc.) and/or where the acceleration mechanisms that cause those energies occur. This will be clarified once the word is put into appropriate context. It is unclear that there is a physical meaning to the particle energy to latitude of peak flux, so perhaps this relation should be omitted. If it is included, it should be demonstrated with a statistical plot of energy versus peak flux latitude, or some such, from the data.

**Reply:** *See answer to comment 6. from Reviewer 1.*

**12.  Comment**

The plume has no relevance to the discussion of the 9-10 MLT hotspot, and should not be included in that discussion. This hotspot has been observed before in the Cattell, Newell and Dombeck et al, papers, for example, as well as tangentially addressed in Frey, and should be discussed within the context.

**Reply:** *The section about low latitudes has been erased.*

**Typo**

Other details that should be addressed include: Page 1, Line 8: The sentence ending with "how" should be reworded.

**Reply:** *This sentence as well as approximately every other sentence in abstract, Section 4 and the summary has been reworded.*

**Typo**

Page 1, Line 13: "main link" should be replaces with something like "a primary link". Solar UV input has much more affect on atmospheric chemistry than particle precipitation, and even affects the precipitation.

**Reply:** *Replaced by "a primary link" as suggested by the reviewer.*

*However we do not agree that UV (generally) has much more affect on atmospheric chemistry than particle precipitation (or why UV radiation should affect the particle precipitation as sugessted by the reviewer).*

*A nice indication of the impact of particle precipitation in comparison to UV radiation is shown in Wissing et al. (2011: Fig. 6). Here the authors present a comparision of the model chain AIMOS (a model for particle precipitation induced atmospheric ionization) and HAMMONIA (a GCM) with the EISCAT incoherent scatter radar. Electron density during night and day, with and without particle precipitation is calculated and compared to radar measurements. The punch line is that even at day-time the UV component just contributes one fifth of the total electon density (at e.g. 110km altitude). Thus particle precipitation is a (or even the) main driver of atmosperic ion chemistry. Of course this comparison covers the auroral region only, but nobody expects particle precipitation to be a dominant driver of atmospheric*

*chemistry where no particles are precipitating. It would be the same as telling that UV radiation is not dominant during night.*

**Typo**

Page 1, Line 16: The statement the auroral particle precipitation *is due to* ... does not make physical sense. This sentence needs to be reworded.

**Reply:** *The sentence now reads: Auroral particle precipitation causes production of $HO_y$ and $NO_x$ and thus is a significant player in mesospheric and stratospheric chemistry, especially as these chemicals catalytically impact the ozone cycle (Callis et al., 1996b, a) and subsequently change the radiation budget and affect dynamics.*

**Typo**

Page 2, Line 22: "over" a wide energy range, rather than "on" a wide energy range.

**Reply:** *Corrected.*

**Typo**

Page 6, Line 25: "Of course", rather than "Of cause". Although that is somewhat colloquial, and isn't really required in the sentence.

**Reply:** *Agreed. The sentence now simply starts with "The SAA ...".*

**Typo**

Page 8, Line 19: "A potential explanation", rather than "An explanation".

**Reply:** *Added a few percent of uncertainty with the word "potential".*

**References**

ASIKAINEN, T. ; MURSULA, K. ; MALINIEMI, V.: Correction of detector noise and recalibration of NOAA/MEPED energetic proton fluxes. In: *J. Geophys. Res.* 117 (2012), S. A09204. `http://dx.doi.org/doi:10.1029/2012JA017593`. – DOI doi:10.1029/2012JA017593.

BORNEBUSCH, J.P. ; WISSING, J.M. ; KALLENRODE, M.-B.: Solar particle precipitation into the polar atmosphere and their dependence on hemisphere and local time. In: *Advances in Space Research* 45 (2010), Nr. 5, 632 - 637. `http://dx.doi.org/https://doi.org/10.1016/j.asr.2009.11.008`. – DOI https://doi.org/10.1016/j.asr.2009.11.008. – ISSN 0273–1177.

CALLIS, L. B. ; BAKER, D. N. ; NATARAJAN, M. ; BERNARD, J. B. ; MEWALDT, R. A. ; SELESNICK, R. S. ; CUMMINGS, J. R.: A 2-D model simulation of downward transport of $NO_y$ into the stratosphere: Effects on the 1994 austral spring $O_3$ and $NO_y$. In: *Geophys. Res. Lett.* 23 (1996), S. 1905–1908. `http://dx.doi.org/10.1029/96GL01788`. – DOI 10.1029/96GL01788.

CALLIS, L. B. ; BOUGHNER, R. E. ; BAKER, D. N. ; MEWALDT, R. A. ; BERNARD BLAKE, J. ; SELESNICK, R. S.

; Cummings, J. R. ; Natarajan, M. ; Mason, G. M. ; Mazur, J. E.: Precipitating electrons: Evidence for effects on mesospheric odd nitrogen. In: *Geophys. Res. Lett.* 23 (1996), S. 1901–1904. `http://dx.doi.org/10.1029/96GL01787`. – DOI 10.1029/96GL01787.

Davis, T. N. ; Sugiura, M.: Auroral electrojet activity index AE and its universal time variations. In: *J. Geophys. Res.* 71 (1966), Februar, S. 785–801. `http://dx.doi.org/10.1029/JZ071i003p00785`. – DOI 10.1029/JZ071i003p00785.

Dombeck, J. ; Cattell, C. ; Prasad, N. ; Meeker, E. ; Hanson, E. ; McFadden, J.: Identification of Auroral Electron Precipitation Mechanism Combinations and Their Relationships to Net Downgoing Energy and Number Flux. In: *Journal of Geophysical Research (Space Physics)* 123 (2018), Dezember, S. 10. `http://dx.doi.org/10.1029/2018JA025749`. – DOI 10.1029/2018JA025749.

Evans, D. S. ; Greer, M. S.: *Polar Orbiting Environmental Satellite Space Environment Monitor - 2, Instrument Descriptions and Archive Data Documentation.* NOAA Space Environ. Lab, Boulder, Colorado, USA: National Oceanic and Atmospheric Administration, 2004. – 156 S. – Version 1.4b, including TED calibrations.

Evans, D. S. ; Greer, M. S.: *Polar Orbiting Environmental Satellite Space Environment Monitor - 2, Instrument Descriptions and Archive Data Documentation.* NOAA Space Environ. Lab, Boulder, Colorado, USA: National Oceanic and Atmospheric Administration, 2006. – 51 S. – Version 2.0.

Gjerloev, J. ; Hoffman, R. ; Friel, M. ; Frank, L. ; Sigwarth, J.: Substorm behavior of the auroral electrojet indices. In: *Annales Geophysicae* 22 (2004), Juni, S. 2135–2149. `http://dx.doi.org/10.5194/angeo-22-2135-2004`. – DOI 10.5194/angeo–22–2135–2004.

Green, Janet: NOAA NESDIS-NGDC MEPED Telescope Data Processing ALGORITHM THEORETICAL BASIS / NOAA National Geophysical Data Center. 2013. – Forschungsbericht. – Version 1.0.

Newell, P. T. ; Gjerloev, J. W.: Evaluation of SuperMAG auroral electrojet indices as indicators of substorms and auroral power. In: *Journal of Geophysical Research (Space Physics)* 116 (2011), Dezember, S. A12211. `http://dx.doi.org/10.1029/2011JA016779`. – DOI 10.1029/2011JA016779.

Newell, P. T. ; Sotirelis, T. ; Wing, S.: Diffuse, monoenergetic, and broadband aurora: The global precipitation budget. In: *Journal of Geophysical Research: Space Physics* 114 (2009), Nr. A9. `http://dx.doi.org/10.1029/2009JA014326`. – DOI 10.1029/2009JA014326.

Newell, Patrick T. ; Lee, Anna R. ; Liou, Kan ; Ohtani, Shin-I. ; Sotirelis, Thomas ; Wing, Simon: Substorm cycle dependence of various types of aurora. In: *Journal of Geophysical Research: Space Physics* 115 (2010), Nr. A9. `http://dx.doi.org/10.1029/2010JA015331`. – DOI 10.1029/2010JA015331.

Reeves, G. D. ; Henderson, M. G. ; Skoug, R. M. ; Thomsen, M. F. ; Borovsky, J. E. ; Funsten, H. O. ; C:son Brandt, P. ; Mitchell, D. J. ; Jahn, J.-M. ; Pollock, C. J. ; McComas, D. J. ; Mende, S. B.: IMAGE, POLAR, and Geosynchronous Observations of Substorm and Ring Current Ion Injection. In: Surjalal Sharma, A. (Hrsg.) ; Kamide, Y. (Hrsg.) ; Lakhina, G. S. (Hrsg.): *Disturbances in Geospace: The Storm-substorm Relationship* Bd. 142, 2003 (Washington DC American Geophysical Union Geophysical Monograph Series), S. 91.

Rodger, Craig J. ; Clilverd, Mark A. ; Green, Janet C. ; Lam, Mai M.: Use of POES SEM-2 observations to examine radiation belt dynamics and energetic electron precipitation into the atmosphere. In: *Journal of Geophysical Research: Space Physics* 115 (2010), Nr. A4. `http://dx.doi.org/10.1029/2008JA014023`. – DOI 10.1029/2008JA014023.

Wissing, J. M. ; Kallenrode, M.-B. ; Kieser, J. ; Schmidt, H. ; Rietveld, M. T. ; Strmme, A. ; Erickson, P. J.: Atmospheric Ionization Module Osnabrck (AIMOS): 3. Comparison of electron density simulations by AIMOS-HAMMONIA and incoherent scatter radar measurements. In: *Journal of Geophysical Research: Space Physics* 116 (2011), Nr. A8. `http://dx.doi.org/10.1029/2010JA016300`. – DOI 10.1029/2010JA016300.

---

## Referee Report (RR1)

**Review of "Magnetic local time asymmetries in precipitating and trapped electron and proton populations with and without substorm activity"**

The authors have put a great deal of work into revising the manuscript and the paper has been significantly improved. My major concern regarding the limited support for the the Kp dependency presented as one of the main findings in the last paper has been thoroughly addressed in this version of the manuscript with the inclusion of several new figures - this is an interesting result.

All of my minor comments and typos have been corrected in the paper or sufficiently argued in the author's reply and I believe that the paper is now in good shape for publication.

---

## Author Response (AR2)

**Response to Reviewer letter**

Olesya Yakovchuk          Jan Maik Wissing

October 13, 2019

Manuscript Title: Magnetic local time asymmetries in precipitating electron and proton populations with and without substorm activity

https://www.ann-geophys-discuss.net/angeo-2019-49

**Reviewer 2 (Comments)**

The paper describes electron and proton differential number flux measured in various energy channels by detectors on POES and METOP from 2001-2008 in the radial direction away from Earth for electrons and 9 degrees from that in protons mapped to 110 km altitude, primarily using the APEX coordinate system and focusing on non-substorm and isolated substorm times, MLT and Kp index dependencies.

The paper is generally well written and presented well (with some exceptions noted below) and will provide useful contributions to the field with some minor modifications. The material is presented with reasonable background and within a reasonable context.

There are no major issues with the paper although there are several minor to mid-level issues that need to be addressed before publication.

In summary, with the relatively minor correction indicated: The conclusions of the paper are of reasonable significance and of importance to the community. The paper length is adequate. The language is fluent and the presentation clear and organized. The figures could be larger but are ok, but the text needs to be made significantly larger, and not at the expense of the size of the data presentation.

**1.   Comment**

First, the abstract lists four findings a), b), c), and d). Strictly speaking neither b nor c are new results. b) (as pointed out later in the paper) is in agreement with conjectures in Newell et al. (2009), and c) has been presented before (although not with a focus on asymmetry) in plots by several authors, at least for some of the energy bands. The consistence of b) with previous results is still a useful finding, so should be included, just reworded. c) also is setup for d) and putting the numerical value on asymmetry is useful. So it is still a useful result, but should be reworded to indicate that it is not a completely new finding.

**Reply:** *The paragraph now reads:*
*Some of the findings are: a) Substorms mostly increase particle precipitation in the night-sector by about factor 24 but can also reduce it in the day-sector. b) MLT dependence can be assigned to particles entering the magnetosphere at the cusp region and magnetospheric particles in combination with energy-specific drifts (in agreement with Newell et al., 2009). c) MLT flux differences of up to two orders of magnitude have been identified inside the auroral oval during geomagnetically disturbed conditions. Novelty here is the comprehensive coverage of energy bands and the focus on asymmetry. d) The maximum flux asymmetry ratio depends on particle energy, decreasing with Kp for low energetic particles and increasing with Kp for higher energy electrons, while high energy protons show a more complex dependency. While some aspects may already been known, the quantification of the flux asymmetry sheds new light on MLT variation.*

**2.   Comment**

The text in all of the figures is dramatically too small.

**Reply:** *The text size has been increased.*

**3.   Comment**

The colors are missing on the color particle flux scale in Figure 2.

**Reply:** *High flux is colored in red, low fluxes in blue and between some fluxes are colored in white - but no colors are missing. Anyhow, the size of the color legend has been increased.*

**4.   Comment**

Text (e.g a paragraph) needs to be added to discuss the limitations of the chosen color scaling in Figures 3 and 4. While the authors choice of scale is appropriate for examining and comparing asymmetry as described. It is atypical and many conclusions that can usually be drawn from color scales in figures do not apply with this choice. For example, the step jumps in the scale creates observable boundaries and structure in some places and plots that may also exist in other places and plots with the same magnitude, but in one case the variations happen to straddle the color boundary, while not in another. For example, the few orange colored pixels about 0 MLT in the upper right (TED4 substorm) plot of Figure 3, seem to indicate higher magnitude and definitive structure, and not having such pixels in the upper left (TED4 non-substorm) plot seems to indicate that structure structure does not exist in that case. But the color scale hard boundaries may be the only cause of these features. Additionally the color magnitudes between plots cannot be directly compared with each other as absolutes or even as relatives. Only the relative color changes/ranges within each given plot is comparable (as well as that variation pattern). The authors use the color scale appropriately and do not make any interpretations outside of the limitations of the color scale, but these limitations need to be expressly stated, at least briefly, to avoid readers from drawing inappropriate meaning from the presented data.

**Reply:** *The scaling is already discussed in detail. The aspect that the reviewer complaines about is color binning. But since any kind of color binning hides or pronounces flux steps (and almost every second paper uses color binning) we do not see why this should be discussed in detail. It is simply a question of accuracy (without color binning) vs. readability (with limited number of color/flux steps).*

**5.   Comment**

A statement needs to be included that indicates how the limitations summed up in the final paragraph of section 3.1 (Page 10, lines 4-7) affect the findings and results of the study. That is, could it changes the findings entirely? Change the magnitude of the findings? or just have minor affects on the results, but not the overall findings and trends?

**Reply:**   *Agreed, this should be clearified. In fact the better satellite coverage in high latitudes is just one part that limits the effect. The main aspect is that no longitudinal variations are seen the lat-long graph. The paragraph now reads:*
*In sum the ascending and descending MLT paths cause the left edge of an equatorial crossing to be affected by the SAA, while the right edge is not. Any MLT analysis of latitudes that show longitudinal variations will suffer from the fact that longitudes contribute very unevenly to the MLT zones. Given that the SAA is the dominant flux source in low latitudes this hampers a MLT flux analysis here. Effects may also be seen in the drift loss cone, where longitudinal flux variations are expected. In high latitudes however, just minor longitudinal variations (in magnetic coordinates) are expected (see Fig. 1, bottom-left, auroral zone). Consequently it just has minor affects on the results, but not the overall findings and trends. Additionally this effect gets counterbalanced by broader MLT coverage and multiple satellites in high latitudes.*

**6.   Comment**

The reference at the end of the second paragraph of section 4.4 (page 15, line 26) should be Figure 14 of Dombeck et al. (2018) rather than Figure 4 of Newell et al. (2009)), and "Alfvén waves" immediately preceding the reference should be changed to "Alfvénic acceleration". There is no need for the authors to go into the depth of literature on Alfvén waves and Alfvénic acceleration of precipitating electrons by the likes of (Keiling, Chaston, Hatch, and Wygant, etc. etc.) But there has been significant work in this area, including many, many maps of Alfvén waves, etc. In short, precipitation is caused by some (and only some) Alfvén waves, and it is then called Alfvénic acceleration. Newell et al. (2009) does not investigate Alfvénic acceleration at all, instead analyzing "broadband" spectra, which Dombeck et al. (2018) indicate is a somewhat/rather imperfect proxy for Alfvénic acceleration. For the purposes of this paper it is better to use the Dombeck et al. (2018) figure reference, as it directly indicates the point the authors are attempting to make, i.e. Alfvénic accelerated precipitation is observed in the mentioned region.

**Reply:** *Both citations are mentioned now.*

**References**

DOMBECK, J. ; CATTELL, C. ; PRASAD, N. ; MEEKER, E. ; HANSON, E. ; MCFADDEN, J.: Identification of Auroral Electron Precipitation Mechanism Combinations and Their Relationships to Net Downgoing Energy and Number Flux. In: *Journal of Geophysical Research (Space Physics)* 123 (2018), Dezember, S. 10. `http://dx.doi.org/10.1029/2018JA025749`. – DOI 10.1029/2018JA025749.

NEWELL, P. T. ; SOTIRELIS, T. ; WING, S.: Diffuse, monoenergetic, and broadband aurora: The global precipitation budget. In: *Journal of Geophysical Research: Space Physics* 114 (2009), Nr. A9. `http://dx.doi.org/10.1029/2009JA014326`. – DOI 10.1029/2009JA014326.